# Multi-Scale Group Relative Policy Optimization for Large Language Models

## Abstract

Reinforcement learning (RL) has become a cornerstone for improving the reasoning ability of large language models (LLMs). The current mainstream Group Relative Policy Optimization (GRPO) estimates advantage via relative comparisons within the full group of sampled responses. However, this single-scale, global comparison mechanism is inherently brittle, sensitive to the heterogeneity and stochasticity of reward distribution, leading to unstable training signals. Drawing inspiration from graph theory, where node importance is better captured through local substructures than global statistics, we propose *Multi-Scale Group Relative Policy Optimization* (MS-GRPO), a novel RL algorithm that generalizes GRPO by aggregating relative advantages computed across multiple response subgroups at varying scales (*e.g.,* pairwise, trios, etc.). Since the exhaustive enumeration of all meaningful subgroups grows combinatorially with group size, we further introduce a practical acceleration scheme that selects a small yet representative subset of subgroups via dilated scale sampling and diversity-aware subgroup selection. In addition, we provide a rigorous theoretical analysis, demonstrating that MS-GRPO can be interpreted as an adaptive correction of GRPO's advantage controlled by the heterogeneity of reward distribution, and gracefully degenerates to GRPO when the reward distribution approaches homogeneity. Experiments demonstrate that MS-GRPO significantly outperforms GRPO on various tasks, for example, with improvements averaged over all evaluated models: +5.5 on AIME24 math reasoning, +4.6 on RiddleSense logical reasoning, +2.7 on Live-CodeBench programming challenges, +2.2 on MedQA medical reasoning, and +13.5 on HotpotQA with search engine.

## 1 Introduction

Large language models (LLMs) have demonstrated unprecedented capabilities in complex reasoning. A key driver behind this success is reinforcement learning (RL), which trains a policy to maximize a reward signal. As a cornerstone algorithm, Proximal Policy Optimization (PPO) (Schulman et al., 2017) suffers from training complexities and instabilities, largely due to its reliance on an online-trained value network for advantage estimation. Recently, Group Relative Policy Optimization (GRPO) (Shao et al., 2024b) emerged as an elegant alternative, which cleverly obviates the need for a learned value function. For each prompt, it samples a group of responses from the current policy model and uses their mean reward as an adaptive baseline. The advantage for each response is then computed by normalizing its reward relative to the group: subtracting the group's mean reward and dividing by the group's standard deviation.

Despite its conceptual simplicity and empirical success, GRPO's advantage estimation mechanism suffers from a fundamental limitation: it performs a single-scale, global comparison across all responses in the group, thereby ignoring the rich, multi-granularity signals embedded in fine-grained, local comparisons. This global normalization is highly sensitive to reward distributional shifts and outlier responses, which are common in practice due to the stochasticity of LLM generation and the occasional instability of reward models. From a graph-theoretic perspective, treating each response as a node in a complete graph, GRPO's approach is equivalent to characterizing each node solely by global graph statistics (mean and variance). However, decades of research in graph analysis have shown that such global characterizations are brittle and limited (Wu et al., 2020; Robinson et al., 2024; Segarra & Ribeiro, 2015; Valente et al., 2008): a node's true importance or role is better captured by its participation in diverse local substructures (*e.g.,* motifs (Milo et al., 2002) and graphlets

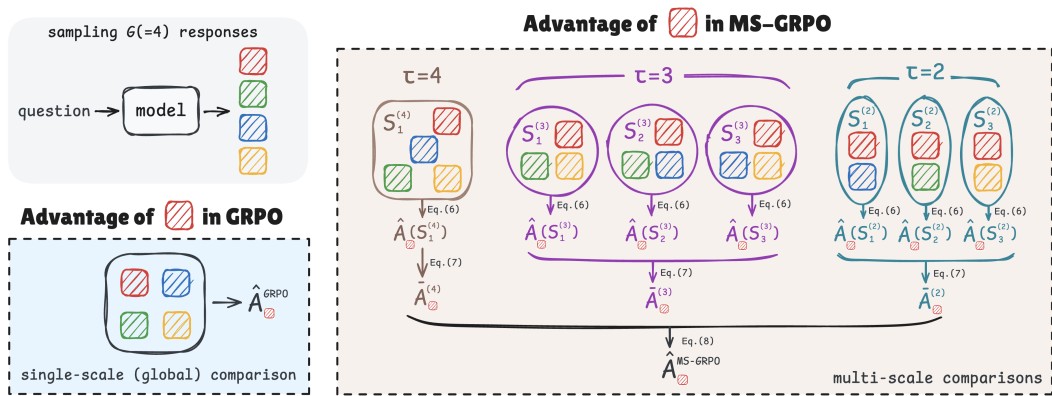

Figure 1: Comparison of the advantage estimation mechanism in MS-GRPO and GRPO. To make an intuitive comparison between them, we only illustrate the advantage of one response (denoted as a red square ▨). GRPO compares ▨ in the full response group, and normalizes its reward using the global mean and standard deviation of this full response group as its relative advantage. In contrast, our MS-GRPO compares ▨ in each valid subgroup containing it, and applies normalization within each comparison subgroup with varying scales (refer to Eq. (4)), and finally synthesizes these multi-view advantages to a holistic advantage via a hierarchical aggregation (refer to Eq. (5) and Eq. (6)).

(Pržulj, 2007)). Analogously, a response's relative advantage should not be judged only against the entire group, but also against various local peer subsets (pairwise, trios, quartets, etc.), each offering a unique perspective on its advantage. Ignoring these multi-scale signals not only reduces the robustness of the advantage estimator but also forfeits valuable information related to the true and reliable value of a response.

In this paper, we propose Multi-Scale Group Relative Policy Optimization (MS-GRPO), a novel RL algorithm that generalizes GRPO by incorporating multi-scale relative comparisons into advantage estimation. Instead of GRPO's single-scale advantage based on global comparison, MS-GRPO constructs a comprehensive advantage signal by aggregating relative advantages computed over all possible subgroups of responses, from pairwise comparisons up to the full group. To ensure statistical fairness, we introduce a hierarchical aggregation strategy that first averages advantages within each scale (*i.e.,* subgroup size) and then combines across scales with tunable weights, preventing larger or more numerous subgroup sizes from dominating the signal.

However, this multi-scale formulation, while conceptually powerful, entails a combinatorial explosion in the number of subgroups as the group size grows, resulting in more computational cost in practice. To address this challenge and enable scalable training, we design a practical **acceleration scheme** (refer to Sec. 3.2) that approximates the full subgroups via two complementary downsampling strategies: (1) *Dilated Scale Sampling*, which sparsely selects a small set of representative scales across the granularity spectrum to reduce redundancy, and (2) *Diverse Group Sampling*, which, for each selected scale, chooses a compact yet maximally diverse subset of subgroups to preserve rich comparative information with minimal redundancy.

In the practical application of RL to LLMs, MS-GRPO demonstrates a notable superiority, stemming from both its solid theoretical foundation and its excellent experimental performance. From a **theoretical standpoint** as detailed in Appendix A, MS-GRPO modifies the advantage estimator of GRPO (Group Relative Policy Optimization) through an adaptive, heterogeneity-aware mechanism. When the reward distribution exhibits high heterogeneity[1], MS-GRPO introduces additional corrections to effectively enhance the reliability of the advantage signal. At its core, MS-GRPO provides an extra advantage boost to samples with above-average rewards while penalizing those that fall below average, thereby improving the signal-to-noise ratio in unstable reward environments. It also places greater emphasis on the raw rewards of samples rather than their normalized relative rewards,

---

[1]The concept "***reward heterogeneity***" refers to the degree of non-uniformity in the reward distribution within the response group. High heterogeneity implies an uneven spread of rewards, often characterized by significant outliers or the emergence of distinct clusters (*e.g.,* high-reward *vs.* low-reward subgroups). This typically arises from randomness of the model's performance or instability of the reward model.

which compensates for the diminished reliability of relative comparisons when the group's reward structure is unstable. When the reward distribution becomes homogeneous, MS-GRPO gracefully degenerates to GRPO, ensuring the method provides robust, context-sensitive advantage estimation when it matters most without introducing unnecessary corrections.

**Experimental results** further corroborate MS-GRPO's superiority. Across multiple tasks aimed at improving LLM reasoning, MS-GRPO consistently outperforms GRPO, delivering performance boosts of +5.5 on AIME24 math reasoning, +4.6 on RiddleSense logical reasoning, +2.7 on Live-CodeBench programming challenges, and +2.2 on MedQA medical reasoning. These improvements are averaged over all evaluated model variants, including the Qwen2.5, LLaMA3.2 and DeepSeek-R1-Distill-Qwen families with 1.5B, 3B, and 7B parameters. Even when LLMs are integrated with search engines, MS-GRPO still significantly outperforms GRPO, achieving gains of +13.5 on HotpotQA benchmark. These experimental data powerfully prove that MS-GRPO can stably and significantly improve LLM performance across various tasks and reward structures.

## 2 BACKGROUND

### 2.1 PRELIMINARY: GROUP RELATIVE POLICY OPTIMIZATION

Group Relative Policy Optimization (GRPO) (Shao et al., 2024b) circumvents the need for value function approximation inherent in PPO (Schulman et al., 2017) by leveraging intra-group comparisons. Specifically, for each question $q$, GRPO samples a group of $G$ responses, $\mathcal{O} = \{o_1, \ldots, o_G\}$ from the old policy model $\pi_{\theta_{\text{old}}}$. Each response $o_i$ consists of a sequence of tokens $(o_{i,1}, \ldots, o_{i,|o_i|})$, with a scalar reward $r_{i,t} \in \mathbb{R}$ assigned to each token $o_{i,t}$. Let $N = \sum_{i=1}^{G} |o_i|$ denote the total number of tokens across all responses in the group. The core of GRPO lies in its advantage estimation. Taking response $o_i$ as an example, it computes an advantage value $\hat{A}_{i,t}^{\text{GRPO}}$ for each token in $o_i$ by normalizing its reward relative to the statistics of the entire group:

$$\hat{A}_{i,t}^{\text{GRPO}} = \frac{r_{i,t} - \mu_{\mathcal{O}}}{\sigma_{\mathcal{O}}}, \tag{1}$$

where $\mu_{\mathcal{O}}$ and $\sigma_{\mathcal{O}}$ are the mean and standard deviation of the rewards within the group $\mathcal{O}$, specifically, $\mu_{\mathcal{O}} = \frac{1}{N} \sum_{i=1}^{G} \sum_{t=1}^{|o_i|} r_{i,t}$ and $\sigma_{\mathcal{O}} = \sqrt{\frac{1}{N} \sum_{i=1}^{G} \sum_{t=1}^{|o_i|} (r_{i,t} - \mu_{\mathcal{O}})^2}$. This advantage $\hat{A}_{i,t}^{\text{GRPO}}$ is subsequently used to optimize the policy model $\pi_{\theta}$ by maximizing the following objective:

$$\mathcal{J}(\theta) = \mathbb{E}_{q \sim \mathcal{D}} \left\{ \frac{1}{G} \sum_{i=1}^{G} \frac{1}{|o_i|} \sum_{t=1}^{|o_i|} [\min(\lambda_{i,t} \hat{A}_{i,t}^{\text{GRPO}}, \texttt{clip}(\lambda_{i,t}, 1-\epsilon, 1+\epsilon) \hat{A}_{i,t}^{\text{GRPO}}) - \beta \, \mathbb{D}_{\text{KL}}(\pi_{\theta} || \pi_{\text{ref}})] \right\}, \tag{2}$$

where $\mathcal{D}$ denotes the training dataset. The importance sampling ratio $\lambda_{i,t} = \frac{\pi_{\theta}(o_{i,t}|q, o_{i,<t})}{\pi_{\theta_{\text{old}}}(o_{i,t}|q, o_{i,<t})}$ corrects for the distributional shift between the behavior old policy model $\pi_{\theta_{\text{old}}}$ that generated the responses and the current policy model $\pi_{\theta}$ being optimized. The $\texttt{clip}(\cdot, 1-\epsilon, 1+\epsilon)$ operation serves to stabilize training by constraining the policy update magnitude, where the clipping bounds are controlled by the hyperparameter $\epsilon$.

### 2.2 MOTIVATION

Although GRPO provides an effective framework for policy alignment, its advantage estimation mechanism is fundamentally limited by its reliance on a single, global comparison across the entire group of $G$ responses, thereby ignoring the rich signals embedded in more localized, fine-grained comparisons. This limitation can be understood through the lens of graph theory. By conceptualizing the responses as nodes in a complete graph ($K_G$), GRPO is tantamount to characterizing each node based solely on the graph's global properties. However, a well-established principle in graph theory is that characterizing a node solely by global graph properties can be misleading and brittle (Wu et al., 2020; Robinson et al., 2024; Bringmann et al., 2019; Robins et al., 2007). Global properties are highly sensitive to the presence of outlier nodes (Segarra & Ribeiro, 2015; Borgatti et al., 2006; Albert et al., 2000; Cavallaro et al., 2024; Žnidaršič et al., 2018) (*e.g.,* responses with exceptionally high or low rewards) and the overall distribution of node attributes (Valente et al., 2008; Karimi et al., 2018; Stoica et al., 2024; Salehzadeh-Yazdi & Hütt, 2025; Martin & Niemeyer, 2021) (*e.g.,* reward distribution of a group of responses). Consequently, GRPO's global comparison baseline is

easily skewed by the abnormal reward noise or specific reward distribution, obscuring a response's more nuanced role within its local neighborhood and yielding an advantage signal that lacks the robustness and precision required for stable policy optimization.

To overcome this limitation of GRPO, we draw inspiration from a key concept in graph analysis: characterizing a node by the ensemble of local subgraphs it participates in, which are often referred to as graphlets (Pržulj, 2007) or network motifs (Milo et al., 2002). In this field, a node's robust identity is defined not by its global position, but by this rich, multi-scale signature of its local environment. Analogously, we propose that a more robust advantage signal can be derived by aggregating a response's relative performance across a diverse set of induced subgraphs of varying scales. For instance, a global comparison might only reveal that a response is above average in the entire group (analogous to a model's average score on broad benchmarks), while our multi-scale approach provides a richer comparison (analogous to a full performance breakdown, including its outstanding performance on some key subfields). We name this paradigm *Multi-Scale Group Relative Policy Optimization (MS-GRPO)* and detail its formulation in the following section.

## 3 METHOD

### 3.1 MULTI-SCALE GROUP RELATIVE POLICY OPTIMIZATION

We propose Multi-Scale Group Relative Policy Optimization (MS-GRPO), whose core innovation lies in a novel advantage estimation mechanism that produces a more robust and reliable advantage by combining the advantages derived from response groups at varying scales. In contrast to GRPO, which performs single-scale advantage estimation via global normalization over the full group of sampled responses for a given question, our MS-GRPO first calculates a set of single-scale advantages by independently normalizing rewards within every valid combination (subset) of responses, ranging from the minimal pairwise combination to the full group. These single-scale advantages at different scales are then fused via a hierarchical aggregation that explicitly balances the contribution of each subset size, mitigating bias stemming from differing numbers of subsets at each size. The resulting multi-scale advantage is used as the advantage signal in the policy optimization objective.

### 3.1.1 MULTI-SCALE ADVANTAGE ESTIMATION

For a given question $q$, MS-GRPO samples a group of $G$ responses $\mathcal{O} = \{o_1, \ldots, o_G\}$ from the old policy model $\pi_{\theta_{\text{old}}}$. Each response $o_i = (o_{i,1}, \ldots, o_{i,|o_i|})$ is a token sequence, with corresponding token-level rewards $\{r_{i,t}\}_{t=1}^{|o_i|}$, where $r_{i,t} \in \mathbb{R}$ is the scalar reward assigned to token $o_{i,t}$. Our multi-scale advantage estimation proceeds in the following three steps:

**(1) Group Construction.** Let's define $\mathbb{S}$ as the $\tau_{\text{min}}$-power-set of the responses $\mathcal{O}$. Different from the normal power set that contains all subsets of $\mathcal{O}$, the $\tau_{\text{min}}$-power-set $\mathbb{S}$ only considers all sub-groups that contain at least $\tau_{\text{min}}$ responses. Formally, we have:

$$\mathbb{S} = \{\mathbb{S}^{(\tau_{\text{min}})}, ..., \mathbb{S}^{(G)}\}, \quad \text{where } \mathbb{S}^{(\tau)} = \{\mathcal{S} \subseteq \mathcal{O} \mid |\mathcal{S}| = \tau\}, \tag{3}$$

where set $\mathbb{S}^{(\tau)}$ is composed of all comparison subgroups with the same scale $\tau$. The hyperparameter $\tau_{\text{min}}$ controlling the minimum comparison scale, with a default value of $\tau_{\text{min}} = 2$, that is, corresponding to pairwise comparison.

**(2) Advantage Estimation.** The core objective of this step is to capture each response's relative advantage within each subgroup. Specifically, for each response $o_i$, we compute its advantages relative to peers in every comparison group containing it, *i.e.*, $\forall_{\mathcal{S} \in \mathbb{S}} o_i \in \mathcal{S}$. We define the advantage assigned to token $o_{i,t}$ within comparison group $\mathcal{S}$ as

$$\hat{A}_{i,t}(\mathcal{S}) = \frac{r_{i,t} - \mu_{\mathcal{S}}}{\sigma_{\mathcal{S}}}, \tag{4}$$

where $\mu_{\mathcal{S}}$ and $\sigma_{\mathcal{S}}$ are the mean and standard deviation of all tokens rewards with in the sub-group $\mathcal{S}$. After this step, each token $o_{i,t}$ is associated with a collection of advantages $\{\hat{A}_{i,t}(\mathcal{S})\}_{\mathcal{S} \in \mathbb{S}, o_i \in \mathcal{S}}$, where each advantage quantifies the token's relative performance within a specific comparison group $\mathcal{S}$, thereby establishing a multi-perspective basis for robust advantage estimation.

**(3) Hierarchical Aggregation.** This step synthesizes the multi-perspective advantages from Step 2 into a comprehensive advantage signal for each token that reflects its holistic relative merit against

the full response group $\mathcal{O}$. Specifically, for each token $o_{i,t}$, we aggregate the advantages $\hat{A}_{i,t}(\mathcal{S})$ across all comparison groups $\mathcal{S}$ containing $o_i$. However, simply averaging over all $\hat{A}_{i,t}(\mathcal{S})$ would introduce a statistical bias due to the unequal number of comparison groups per size. For response $o_i$, the number of comparison groups of size $\tau$ containing $o_i$ is $\binom{G-1}{\tau-1}$, since $o_i$ is fixed to be included and the remaining $\tau - 1$ responses are chosen from the other $G - 1$ responses in the full group $\mathcal{O}$. This quantity peaks near $\tau = G/2$. As a result, medium-sized comparison groups would dominate the aggregation, not because they provide higher-quality signals, but solely because they are more numerous, leading to a biased advantage estimation. To avoid such bias, we perform a hierarchical aggregation as follows: **(1)** For each token $o_{i,t}$, we compute the scale-specific averaged advantage at each scale $\tau \in \{\tau_{\min}, \dots, G\}$:

$$\bar{A}_{i,t}^{(\tau)} = \frac{1}{\binom{G-1}{\tau-1}} \sum_{\substack{\mathcal{S} \in \mathbb{S} \\ |\mathcal{S}| = \tau, \, o_i \in \mathcal{S}}} \hat{A}_{i,t}(\mathcal{S}), \tag{5}$$

which averages all advantages from comparison groups of size $\tau$ that contain $o_i$, yielding an averaged advantage at a specific scale. **(2)** We combine these scale-specific advantages into a holistic multi-scale advantage:

$$\hat{A}_{i,t}^{\text{MS-GRPO}} = \sum_{\tau=\tau_{\min}}^{G} w_\tau \cdot \bar{A}_{i,t}^{(\tau)}, \quad \text{where } w_\tau \geq 0 \text{ and } \sum_{\tau=\tau_{\min}}^{G} w_\tau = 1. \tag{6}$$

The weight coefficient $w_\tau$ is a hyperparameter controlling the contribution of scale $\tau$. Uniform weights ($w_\tau = \frac{1}{G-\tau_{\min}+1}$) are used by default, treating all scales equally. Alternatively, manually-designed weighting schemes can also be flexibly implemented, such as assigning higher weights to larger scales (corresponding to larger comparison groups).

### 3.1.2 TRAINING OBJECTIVE

Building on our multi-scale advantage $\hat{A}_{i,t}^{\text{MS-GRPO}}$ from Eq. (6), policy optimization proceeds by adopting the GRPO-style objective defined in Eq. (2), with the sole modification that the original advantage term $\hat{A}_{i,t}^{\text{GRPO}}$ is replaced with our more reliable multi-scale advantage $\hat{A}_{i,t}^{\text{MS-GRPO}}$. This substitution enables more reliable policy updates by leveraging multi-scale relative comparisons.

## 3.2 PRACTICAL ACCELERATION SCHEME

Although the multi-scale advantage estimation of MS-GRPO described in Sec. 3.1.1 is theoretically sound, its computational complexity presents a practical scalability challenge. The total number of valid comparison groups (*i.e.,* the size of $\mathbb{S}$ defined in Eq. (3)) grows exponentially with the group size $G$, rendering the computation of the multi-scale advantage computationally intractable for large group size $G$.

To ensure the scalability of MS-GRPO, we introduce a practical acceleration scheme based on approximation via subsampling. The core idea is to compute an approximated multi-scale advantage by operating not on the exhaustive set $\mathbb{S}$, but on a much smaller, representative subset $\mathbb{C} \subset \mathbb{S}$. The construction of this subset $\mathbb{C}$ is approached along two orthogonal dimensions to reduce redundancy. First, we apply a **Dilated Scale Sampling** strategy to select a representative but sparse set of scales, denoted as $\mathbb{T}$. Second, for each scale $\tau \in \mathbb{T}$, we employ a **Diverse Group Sampling** procedure to select a concise yet informative subset of comparison groups, denoted as $\mathbb{C}^{(\tau)}$. The final, overall representative subset $\mathbb{C}$ is then formed by the union of these per-scale subsets $\mathbb{C} = \cup_{\tau \in \mathbb{T}} \mathbb{C}^{(\tau)}$. The advantage estimation process (Steps (2) and (3) in Sec. 3.1.1) is then performed exclusively on this condensed set $\mathbb{C}$. The specific mechanisms for each of these two sampling dimensions are detailed in the following Sec. 3.2.1 and Sec. 3.2.2, respectively.

### 3.2.1 DILATED SCALE SAMPLING FOR SCALE SELECTION

In MS-GRPO, a *scale* refers to the size $\tau = |\mathcal{C}|$ of a comparison group, which determines the granularity of the relative comparison. Small scales (*e.g.,* $\tau = 2$) capture fine-grained, local comparisons, while large scales (*e.g.,* $\tau = G$) reflect coarse-grained, global comparisons. A full estimation that considers all consecutive scales inherently introduces redundancy, as adjacent scales (*e.g.,* $\tau$ and $\tau + 1$) yield highly correlated advantage signals.

Inspired by the classical dilated convolution (Yu & Koltun, 2015), we perform analogous dilated sampling over the scale dimension to ensure balanced coverage across the entire scale spectrum. Specifically, given a hyperparameter $M$ representing the target number of scales, our *Dilated Scale Sampling* strategy partitions the full scale range $[\tau_{\min}, G]$ into $M$ non-overlapping intervals and selects one scale per interval. This process is formalized in two steps: **First**, we partition the integer range $[\tau_{\min}, G]$ into $M$ contiguous and non-overlapping intervals, $\{I_j\}_{j=1}^M$. The $j$-th interval, $I_j = [s_j, e_j]$, is formally defined by its start point $s_j = \max(\tau_{\min}, e_{j-1} + 1)$ and end point $e_j = G - (M - j) \cdot \left\lceil \frac{G - \tau_{\min} + 1}{M} \right\rceil$. **Second**, we uniformly sample one scale $\tau_j \sim \mathrm{Uniform}(I_j)$ from each interval. This process yields the final set of selected scales $\mathbb{T} = \{\tau_1, \ldots, \tau_M\}$, which forms the basis for the per-scale group sampling described next.

### 3.2.2 DIVERSE GROUP SAMPLING FOR PER-SCALE COMPARISON GROUP SELECTION

For each scale $\tau \in \mathbb{T}$ selected above, the number of possible comparison groups is $\binom{G}{\tau}$, which can be prohibitively large and informationally redundant. For instance, two groups that differ by only one member provide highly correlated advantage signals, and averaging over them yields diminishing returns. Our goal, therefore, is to select a concise subset of comparison groups that maximizes internal diversity. To achieve this, we apply a conditional sampling strategy controlled by a budget hyperparameter $K$. Let $\mathbb{S}^{(\tau)} = \{\mathcal{S} \subseteq \mathcal{O} \mid |\mathcal{S}| = \tau\}$ denote the set of all valid comparison groups at scale $\tau$, and $N_\tau = |\mathbb{S}^{(\tau)}| = \binom{G}{\tau}$. We define the representative subset $\mathbb{C}^{(\tau)}$ as follows:

$$\mathbb{C}^{(\tau)} = \begin{cases} \mathbb{S}^{(\tau)} & \text{if } N_\tau \leq K, \\ K \text{ diverse comparison groups from } \mathbb{S}^{(\tau)} & \text{otherwise.} \end{cases} \quad (7)$$

In the latter case, we formalize the selection as a *diversity maximization optimization problem*. Specifically, we aim to find a subset $\mathbb{C}^{(\tau)} \subset \mathbb{S}^{(\tau)}$ of size $K$ that maximizes the sum of pairwise Jaccard distances $d_J$ between its members:

$$\mathbb{C}^{(\tau)*} = \underset{\mathbb{C}^{(\tau)} \subset \mathbb{S}^{(\tau)}, |\mathbb{C}^{(\tau)}| = K}{\arg\max} \sum_{\mathcal{S}_a, \mathcal{S}_b \in \mathbb{C}^{(\tau)}, a \neq b} d_J(\mathcal{S}_a, \mathcal{S}_b), \quad (8)$$

where $d_J(\mathcal{S}_a, \mathcal{S}_b) = 1 - \frac{|\mathcal{S}_a \cap \mathcal{S}_b|}{|\mathcal{S}_a \cup \mathcal{S}_b|}$ defines the divergence between any two groups $\mathcal{S}_a, \mathcal{S}_b \in \mathbb{S}^{(\tau)}$. As this problem is NP-hard, we design a fast, polynomial-time greedy algorithm that offers theoretical approximation guarantees (derived in Appendix E). Specifically, starting with an empty set $\mathbb{C}^{(\tau)} = \emptyset$, the algorithm iteratively constructs the set $\mathbb{C}^{(\tau)}$ by repeatedly adding the candidate group $\mathcal{S}^*$ from the remaining pool $(\mathbb{S}^{(\tau)} \setminus \mathbb{C}^{(\tau)})$ that is most dissimilar to the groups already selected:

$$\mathcal{S}^* = \underset{\mathcal{S} \in \mathbb{S}^{(\tau)} \setminus \mathbb{C}^{(\tau)}}{\arg\max} \sum_{\mathcal{S}' \in \mathbb{C}^{(\tau)}} d_J(\mathcal{S}, \mathcal{S}'). \quad (9)$$

This greedy process terminates after $K$ iterations, yielding the final representative set $\mathbb{C}^{(\tau)}$ of size $K$. By construction, for each selected scale $\tau \in \mathbb{T}$, we have $|\mathbb{C}^{(\tau)}| \leq K$. Since $|\mathbb{T}| = M$, the total number of comparison groups used in the accelerated MS-GRPO is bounded by $M \cdot K$, ensuring computational tractability without sacrificing representativeness.

## 4 EXPERIMENTS

In this section, we first compare the performance of our MS-GRPO and GRPO on a wide range of tasks in Sec. 4.1, and then ablate each of our key designs in Sec. 4.2. The experimental settings are detailed in Appendix B.

### 4.1 MAIN RESULTS

**Math Reasoning.** As shown in Table 1, our MS-GRPO demonstrates comprehensive superiority on all math reasoning benchmarks, significantly and consistently outperforming GRPO across both the Qwen and LLaMA series models. For example, it achieves additional average accuracy gains of +4.1 for Qwen2.5-Math-7B and +5.0 for LLaMA3.2-3B-Instruct over GRPO across the five benchmarks. It is worth noting that as the model size increases (from 1.5B to 7B), the improvement brought by our

Table 1: Comparison with GRPO on five challenging math reasoning benchmarks.

| Model | AIME24 | AMC23 | MATH-500 | MinervaMath | OlympiadBench | Avg. |
|---|---|---|---|---|---|---|
| Qwen2.5-Math-1.5B | 13.3 | 30.0 | 36.6 | 19.1 | 22.6 | 24.3 |
| + GRPO | 20.0 | 57.5 | 76.4 | 32.3 | 38.5 | 44.9 |
| + MS-GRPO (Ours) | 26.7 (+6.7) | 62.5 (+5.0) | 80.0 (+3.6) | 33.4 (+1.1) | 40.5 (+2.0) | 48.6 (+3.7) |
| Qwen2.5-Math-7B | 13.3 | 40.0 | 53.6 | 17.2 | 17.4 | 28.3 |
| + GRPO | 33.3 | 67.5 | 82.0 | 36.0 | 42.6 | 52.3 |
| + MS-GRPO (Ours) | 40.0 (+6.7) | 72.5 (+5.0) | 82.8 (+0.8) | 40.4 (+4.4) | 46.4 (+3.8) | 56.4 (+4.1) |
| Qwen2.5-Math-1.5B-Instruct | 10.0 | 60.0 | 74.2 | 32.3 | 39.5 | 43.2 |
| + GRPO | 16.7 | 62.5 | 76.4 | 30.8 | 40.1 | 45.3 |
| + MS-GRPO (Ours) | 20.0 (+3.3) | 62.5 (+0.0) | 77.6 (+1.2) | 33.0 (+2.2) | 41.7 (+1.6) | 47.0 (+1.7) |
| Qwen2.5-Math-7B-Instruct | 13.3 | 70.0 | 81.2 | 36.0 | 45.6 | 49.2 |
| + GRPO | 16.7 | 70.0 | 82.6 | 38.7 | 46.8 | 51.0 |
| + MS-GRPO (Ours) | 23.3 (+6.6) | 75.0 (+5.0) | 83.5 (+0.9) | 40.4 (+1.7) | 47.8 (+1.0) | 54.0 (+3.0) |
| LLaMA3.2-3B-Instruct | 3.3 | 22.5 | 48.0 | 16.5 | 14.5 | 21.0 |
| + GRPO | 13.3 | 27.5 | 57.2 | 20.9 | 21.6 | 28.1 |
| + MS-GRPO (Ours) | 20.0 (+6.7) | 40.0 (+12.5) | 59.6 (+2.4) | 22.7 (+1.8) | 23.2 (+1.6) | 33.1 (+5.0) |
| DeepSeek-R1-Distill-Qwen-1.5B | 28.8 | 62.9 | 82.8 | 26.5 | 43.3 | 48.9 |
| + GRPO | 30.0 | 67.5 | 83.8 | 29.7 | 47.0 | 51.6 |
| + MS-GRPO (Ours) | 33.2 (+3.2) | 75.0 (+7.5) | 86.0 (+2.2) | 31.1 (+1.4) | 49.4 (+2.4) | 54.9 (+3.3) |

Table 2: Comparison with GRPO on five code generation benchmarks.

| Model | LiveCodeBench | HumanEval | HumanEval+ | MBPP | MBPP+ | Avg. |
|---|---|---|---|---|---|---|
| Qwen2.5-7B-Instruct | 22.4 | 86.4 | 80.5 | 75.6 | 66.7 | 66.3 |
| + GRPO | 28.6 | 87.8 | 84.0 | 80.4 | 68.9 | 70.0 |
| + MS-GRPO (Ours) | 30.6 (+2.0) | 88.5 (+0.7) | 84.8 (+0.8) | 82.5 (+2.1) | 70.8 (+1.9) | 71.4 (+1.4) |
| Qwen2.5-Coder-7B-Instruct | 30.7 | 86.0 | 83.2 | 82.5 | 69.7 | 70.4 |
| + GRPO | 32.6 | 87.1 | 83.3 | 83.7 | 70.1 | 71.4 |
| + MS-GRPO (Ours) | 36.0 (+3.4) | 88.0 (+0.9) | 84.2 (+0.9) | 85.9 (+2.2) | 74.4 (+4.3) | 73.7 (+2.3) |

MS-GRPO also grows instead of narrows, which highlights the excellent scalability of our method. Furthermore, our method exhibits a more pronounced superiority on more difficult benchmarks, such as AIME24. These results provide strong evidence that, compared to GRPO, our proposed MS-GRPO can more effectively unlock the deep reasoning potential of LLMs for solving complex mathematical problems.

**Code Reasoning.** For code generation, MS-GRPO showcases similarly strong performance. As shown in Table 2, MS-GRPO delivers consistent performance gains over GRPO across all five code benchmarks. In particular, MS-GRPO achieves higher advantage of the code-specialized model, Qwen2.5-Coder-7B-Instruct, than that of the general model, Qwen2.5-7B-Instruct (+2.3 *vs.* +1.4). Similar with the findings in math reasoning, the gains of MS-GRPO relative to GPRO becomes more substantial on challenging benchmarks, for example, +3.4 on LiveCodeBench and +4.3 on MBPP+. This clearly indicates that MS-GRPO not only excels at general-purpose code tasks but also effectively enhances LLMs' ability to tackle complex programming challenges.

**Logical Reasoning.** Logical reasoning, often assessed with puzzles, is a key indicator of the intelligence of LLMs. We evaluate on the well-known RiddleSense benchmark. The results in Table 3 clearly demonstrate the significant superiority of MS-GRPO compared with GRPO. On the smaller 1.5B level models, MS-GRPO surpasses GRPO by +5.3 and +5.9, respectively. On the larger 7B models, the advantage remains notable at +4.0 and +3.3. It shows that MS-GRPO can effectively unlock and enhance the logical reasoning capabilities of models, particularly for smaller models.

**Medical Reasoning.** In the highly specialized medical domain, MS-GRPO continues to demonstrate its superiority over GRPO. As shown in Table 4, on the authoritative MedQA benchmark (based on the US Medical Licensing Examination), our MS-GRPO consistently outperforms GRPO with gains of +2.0 to +2.5 across all baseline LLMs. This result validates the generalizability and effectiveness of MS-GRPO in specialized domains, showcasing its power to further enhance LLMs' ability on tasks requiring deep, domain-specific knowledge.

**Question Answering with Search Engine.** We also compare MS-GRPO with GRPO for training LLMs integrated with a search engine. In this setting, the model can query a search engine to retrieve relevant external knowledge to aid its reasoning process. Therefore, this training setting is particularly challenging since it requires enhancing not only the model's core reasoning abilities but

Table 3: Comparison with GRPO on logical reasoning benchmark (RiddleSense).

| Model | before training | + GRPO | + MS-GRPO (Ours) |
|---|---|---|---|
| Qwen2.5-1.5B | 6.4 | 65.0 | 70.3 (+5.3) |
| Qwen2.5-7B | 60.2 | 73.8 | 77.8 (+4.0) |
| Qwen2.5-1.5B-Instruct | 36.5 | 64.3 | 70.2 (+5.9) |
| Qwen2.5-7B-Instruct | 65.5 | 76.0 | 79.3 (+3.3) |

Table 4: Comparison with GRPO on medical reasoning benchmark (MedQA).

| Model | before training | + GRPO | + MS-GRPO (Ours) |
|---|---|---|---|
| Qwen2.5-1.5B | 8.4 | 42.2 | 44.2 (+2.0) |
| Qwen2.5-7B | 47.8 | 61.3 | 63.4 (+2.1) |
| Qwen2.5-1.5B-Instruct | 19.4 | 42.9 | 45.0 (+2.1) |
| Qwen2.5-7B-Instruct | 60.5 | 64.1 | 66.6 (+2.5) |

Table 5: Comparison with GRPO on general and multi-hop QA tasks for LLMs with search engine.

| Benchmarks | Qwen2.5-1.5B | | | Qwen2.5-3B | | |
|---|---|---|---|---|---|---|
| | before training | + GRPO | + MS-GRPO (Ours) | before training | + GRPO | + MS-GRPO (Ours) |
| NQ | 0.2 | 19.4 | 38.4 (+19.0) | 2.3 | 40.6 | 44.1 (+3.5) |
| HotpotQA | 0.4 | 18.3 | 30.8 (+12.5) | 2.1 | 28.4 | 42.8 (+14.4) |
| Avg. | 0.3 | 18.9 | 34.6 (+15.7) | 2.2 | 34.5 | 43.5 (+9.0) |

also its proficiency in interacting with the search engine. As shown in Table 5, MS-GRPO comprehensively and significantly outperforms GRPO on both general QA benchmark (NQ) and multi-hop QA benchmark (HotpotQA), with the average bonus of +15.7 and +9.0 over GRPO for Qwen2.5-1.5B and Qwen2.5-3B, respectively. Notably, MS-GRPO is exceptionally effective at unlocking the potential of smaller models. Specifically, for Qwen2.5-1.5B, it delivers an impressive average performance boost of +15.7, and achieves a massive gain of +19.0 on NQ benchmark, nearly doubling the performance of GRPO. Furthermore, for Qwen2.5-3B, MS-GRPO shows larger improvements on the more challenging HotpotQA than NQ, achieving remarkable gains of +12.5 and +14.4 over GRPO. These results clearly indicate that MS-GRPO not only enhances a model's intrinsic reasoning ability but also significantly optimizes its performance when acting as an agent that interacts with external tools like a search engine.

## 4.2 ABLATION STUDY

We ablate the efficacy of two key components in MS-GRPO: (1) the *Hierarchical Aggregation* strategy for multi-scale advantages, and (2) the *Practical Acceleration Scheme* for scalable computation. Ablation experiment on each term is conducted under two training settings: training LLaMA3.2-3B-Instruct on math dataset, and training Qwen2.5-Coder-7B-Instruct on code dataset. We report the average pass@1 across five benchmarks for math and code evaluation, respectively.

**Efficacy of Hierarchical Aggregation.** Hierarchical aggregation (detailed in Sec. 3.1) is a core component of MS-GRPO that first averages advantages within each scale and then combines across scales with weighted summation, making a fair contribution for all scales. We investigate two aspects of this design:

- *Necessity:* To evaluate its necessity, we compare it against a naive aggregation that directly averages all per-subgroup advantages without scale-wise awareness. As shown in Table 6a (line 1-2), this naive manner underperforms hierarchical aggregation by 1.5 on math and 0.8 on code, confirming that explicit scale-wise averaging is crucial to avoid bias from combinatorial imbalance.
- *Impact of weighting scheme* $\{w_\tau\}$: We compare three different weight coefficients $\{w_\tau\}_\tau$ in Eq. (6): (1) *uniform*: equal weight per scale, (2) *global-biased*: higher weight for larger scales, emphasizing global comparisons. (3) *local-biased*: higher weight on smaller scales, emphasizing local comparisons. In all cases, the weights are normalized to sum to one. As Table 6a (line 2-4), uniform weighting achieves the best performance, while the local-biased weighting performs worst. Notably, even the global-biased variant outperforms GRPO by +1.4 and +2.0, demonstrating that incorporating any form of multi-scale signal can improve upon GRPO's single-scale advantage estimation.

**Efficacy of Acceleration Scheme.** Our acceleration scheme (detailed in Sec. 3.2) consists of two orthogonal components: Dilated Scale Sampling and Diverse Group Sampling. We analyze their individual and combined effects below:

Table 6: Ablation study on two key designs of MS-GRPO: (a) Hierarchical Aggregation (HA) and its weighting scheme; (b) Practical Acceleration scheme with two components, Dilated Scale Sampling (DSS) and Diverse Group Sampling (DGS). The impact of hyperparameters $M$ and $K$ is evaluated in (c) and (d), respectively, where the reported scores follow the format *math / code* performance. The light blue shallow highlighted our default configuration. For reference, GRPO achieves 28.1 (math) and 71.4 (code) in average `pass@1` (%), with a training time of 184 seconds (s) per step.

(a)

| HA | weighting scheme | Math (Avg) | Code (Avg) |
|---|---|---|---|
| ✗ | - | 31.6 | 72.9 |
| ✓ | uniform | **33.1** | **73.7** |
| ✓ | global-biased | 32.7 | 73.4 |
| ✓ | local-biased | 31.3 | 72.5 |

(b)

| DSS | DGS | Math (Avg) | Code (Avg) | Time per step (Avg) | Comparison Group Number |
|---|---|---|---|---|---|
| ✗ | ✗ | **33.3** | **74.0** | 233s | 247 |
| ✓ | ✗ | **33.3** | **74.0** | 220s | 93~155 |
| ✗ | ✓ | 33.1 | 73.8 | 203s | 49 |
| ✓ | ✓ | 33.1 | 73.7 | 199s | 25 |

(c)

| DSS sampling mode | M | | | no sampling |
|---|---|---|---|---|
| | 2 | 4 | 6 | |
| fixed | 30.4 / 72.0 | 31.5 / 73.2 | 32.9 / 73.4 | 33.1 / 73.8 |
| random | 31.6 / 72.7 | **33.1 / 73.7** | 33.1 / 73.7 | |

(d)

| DGS sampling mode | K | | | | no sampling |
|---|---|---|---|---|---|
| | 4 | 8 | 16 | 32 | |
| random | 32.6 / 72.7 | 32.6 / 72.9 | 32.9 / 73.5 | 33.1 / 73.9 | **33.3 / 74.0** |
| optimized | 32.7 / 73.2 | 33.1 / 73.7 | 33.0 / 73.7 | **33.3** / 73.9 | |

- ***Impact of Dilated Scale Sampling (DSS):*** DSS reduces redundancy across scales by selecting only $M$ representative scales from the full range $[\tau_{\min}, G]$. As shown in Table 6b, applying DSS only reduces the number of comparison groups by nearly half while maintaining the performance (line 1-2). A key design choice in DSS is the sampling strategy. Instead of deterministically selecting scales at fixed intervals (*e.g.,* $\tau = 2, 4, 6, 8$), we partition the scale range into $M$ contiguous intervals and randomly sample one scale per interval. As Table 6c shows, this random sampling strategy consistently outperforms fixed-interval sampling across all $M$ values. Therefore, random sampling within each interval is the merit of maintaining the performance during scale sparsification. We attribute this gain to the stochasticity introduced during training: random scale selection ensures that all scales have a chance to be used over time, reducing human-designed bias in fixed-interval sampling and improving generalization. Table 6c further studies the sensitivity to hyperparameter $M$, the number of scales to be retained. Performance improves as $M$ increases, but with diminishing returns beyond $M = 4$. Given the trade-off between efficiency and performance, we adopt $M = 4$ as the default.

- ***Impact of Diverse Group Sampling (DGS):*** DGS controls the number of comparison groups per scale via a budget $K$. As shown in Table 6b and Table 6d, disabling DGS (*i.e.,* using all subgroups) yields the best performance, while enabling DGS with $K = 8$ slightly reduces scores (from 33.3 to 33.1 on math, 74.0 to 73.7 on code), a minor drop given the drastic reduction in the number of comparison groups (from 247 to 25). Crucially, the *diversity-aware selection* in DGS is essential. Table 6d shows that the optimized strategy consistently outperforms random sampling, especially at small $K$, indicating that maximizing subgroup diversity preserves more informative signals under tight budgets $K$. Finally, we analyze the effect of $K$. Performance increases with $K$ and saturates around $K = 16$ or $32$. However, even with $K = 8$, our method achieves 99.4% of the full performance (33.1 vs. 33.3 on math), thus, we choose $K = 8$ as the default, striking an optimal balance between efficiency and signal fidelity.

## 5 CONCLUSION

We propose Multi-Scale Group Relative Policy Optimization (MS-GRPO), which improves upon GRPO by leveraging multi-scale comparisons across response subgroups to generate more robust and reliable advantage signals. MS-GRPO mitigates the brittleness of global normalization under reward heterogeneity and stochasticity. We further introduce a practical acceleration scheme to ensure its efficiency and scalability.

**LLM-Usage Statement.** The authors used a large language model to assist with language polishing, grammar correction, and typo identification in this paper. The ideas, methodology, experimental design, and results presented are the sole work of the authors.

ETHICS STATEMENT

Based on the ICLR Code of Ethics, we confirm that our research adheres to its principles. The primary contribution of our work is to improve the training effectiveness, efficiency, and generalization performance of large language models during the Reinforcement Learning from Human Feedback (RLHF) and fine-tuning stages. We believe these advancements will allow large models to better serve society. We acknowledge the importance of the responsible application of this technology. Our research does not involve the collection or use of any new personally identifiable information, and all experiments were conducted on publicly available datasets.

REPRODUCIBILITY STATEMENT

We are fully committed to ensuring that our research is reproducible. To this end, we've included all the essential details required for independent verification, including comprehensive descriptions of our model architectures, training procedures, and all relevant hyperparameter settings in the main text and supplementary materials. For complete transparency and to facilitate future research, we plan to make the source code and all associated datasets publicly available as soon as this paper is officially accepted and published.

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

# A    ANALYSIS ON THE ADVANTAGE DIFFERENCE OF MS-GRPO AND GRPO

To further understand the distinction between the advantage estimators of MS-GRPO and GRPO, we derive an analytical expression for their difference, denoted as $\Delta \hat{A}_{i,t}$. The complete derivation is provided in Appendix D.1, yielding the following formula (where $\mathrm{Var}(\cdot)$ denotes the variance):

$$
\Delta \hat{A}_{i,t} = \hat{A}_{i,t}^{\text{MS-GRPO}} - \hat{A}_{i,t}^{\text{GRPO}}
$$

$$
\approx \underbrace{\underbrace{(r_{i,t} - \mu_{\mathcal{O}})}_{\text{global-relative reward}} \overbrace{\left( \frac{1}{\mathbb{E}_{\mathcal{S}}[\sigma_{\mathcal{S}}]} - \frac{1}{\sigma_{\mathcal{O}}} \right)}^{\text{Correction Term 1}}}_{\text{scaling factor 1}} + \underbrace{\underbrace{r_{i,t}}_{\text{reward}} \overbrace{\frac{\mathrm{Var}_{\mathcal{S}}(\sigma_{\mathcal{S}})}{(\mathbb{E}_{\mathcal{S}}[\sigma_{\mathcal{S}}])^3}}^{\text{Correction Term 2}}}_{\text{scaling factor 2}} \tag{10}
$$

This difference can be interpreted as an additive correction bias that MS-GRPO applies to the GRPO's advantage. It consists of two distinct correction terms: the first scales the *global-relative reward* $(r_{i,t} - \mu_{\mathcal{O}})$, while the second scales the *raw reward* $(r_{i,t})$.

***Rationale of Correction Term 1.*** The first correction term can be understood as using the difference between each token's reward $r_{i,t}$ and the global average reward $\mu_{\mathcal{O}}$ to correct its relative advantage. Since "scaling factor 1 $(1/\mathbb{E}_{\mathcal{S}}[\sigma_{\mathcal{S}}] - 1/\sigma_{\mathcal{O}})$" is non-negative (proven in Appendix Proof 1), the sign of "correction term 1" is determined solely by whether the token's reward exceeds or falls below the global average reward. Among all tokens in the G responses, it grants an additional advantage bonus to tokens above the global reward level, while imposing an additional advantage penalty to tokens below the global reward level. In addition to being affected by the absolute deviation $|r_{i,t} - \mu_{\mathcal{O}}|$, the magnitude of these additional advantage bonus or penalties also increases with the heterogeneity of the reward distribution, since the magnitude of the "scaling factor 1" increases with the heterogeneity of the reward distribution (proven in Appendix Proof 2). This is reasonable because if the model's answer quality to the current question is unstable (high heterogeneity), a response that significantly outperforms the average is more notable and thus deserves a larger advantage boost.

***Rationale of Correction Term 2.*** The second correction term can be understood as using the scaled reward value of each token to correct its relative advantage. Since "scaling factor 2" is non-negative (proven in Appendix Proof 3), the sign of the second correction term is determined by the positive or negative value of the raw reward $r_{i,t}$. Among all tokens in the $G$ responses, this term assigns a higher additional advantage boost for higher-quality tokens (corresponding to higher rewards). In addition to being affected by the reward value of the token, the magnitude of this correction term is also affected by the reward heterogeneity. The greater the heterogeneity, the larger the magnitude of the second correction term, because the magnitude of "scaling factor 2" increases with the heterogeneity of the reward distribution (proven in Appendix Proof 4). This is reasonable because if the reward distribution is highly heterogeneous (*e.g.,* caused by the instability of model performance or reward model), the reward groups for advantage estimation will also be unstable, and the reliability of relative comparisons will be weakened. Therefore, in this highly uncertain comparison environment, it is reasonable to place slightly more trust in a token's absolute superiority (reflected by its raw reward $r_{i,t}$), as the second correction term does.

**Overall Interpretation: MS-GRPO as an Adaptive Advantage Correction.** In summary, our analysis reveals that the advantage difference $\Delta \hat{A}_{i,t}$ acts as an adaptive, heterogeneity-aware correction that MS-GRPO applies to the GRPO's advantage. The behavior of this correction is dictated by the structure of the reward distribution:

- When the reward distribution is nearly *homogeneous*, the two variance related to the reward heterogeneity ($\mathrm{Var}_{\mathcal{S}}(\mu_{\mathcal{S}})$ and $\mathrm{Var}_{\mathcal{S}}(\sigma_{\mathcal{S}})$) approach zero. Consequently, both "scaling factor 1" and "scaling factor 2" also approach zero, causing the entire advantage difference $\Delta \hat{A}_{i,t}$ to vanish. In this simple case, MS-GRPO gracefully degenerates to GRPO, demonstrating that the correction is adaptively applied only when necessary.
- Conversely, when the reward distribution is highly *heterogeneous*, the correction term becomes more significant. The correction operates through two synergistic mechanisms. "Correction Term 1" grants an additional advantage boost to tokens with above-average rewards (since outperforming the average is more significant and notable in an unstable sampling), and imposes an additional advantage penalty on those below average. Simultaneously, "Correction Term 2" places greater trust in a token's raw reward when the high heterogeneity makes the context

for relative comparisons unstable. Both terms, therefore, leverage the rich, fine-grained information embedded in the group's reward heterogeneity to produce a more nuanced and reliable advantage signal. This stands in sharp contrast to GRPO's single-scale, global estimation, which completely overlooks the crucial structural information.

The combined effect results in a sophisticated correction that not only accounts for a token's relative standing but also the reliability of the context in which that standing is measured, leading to a more robust and nuanced advantage signal.

## B  EXPERIMENTAL SETTINGS

**Implementation Details.** Our implementation is built upon the VeRL (Sheng et al., 2025) library. For the hierarchical aggregation in Eq. (6), we adopt uniform weights across all scales (*i.e.,* $w_\tau = 1/(G - \tau_{\min} + 1)$) by default, because we empirically find that this simple setting consistently yields strong performance. In fact, linearly assigning higher weights to larger scales underperforms uniform weighting, as verified in our ablation study (see Table 4.2 (a)). For the two hyperparameters introduced in our acceleration scheme, we set $M = 4$ and $K = 8$ by default in our experiments. These values strike a favorable trade-off between computational efficiency and advantage estimation fidelity across all tasks, and we ablate the effect of them in Table 6c and Table 6d.

**Training Settings.** For each training data, we sample $G = 8$ responses from the current policy with temperature 1.0. The maximum output sequence length is set as 2048 tokens. We use a constant learning rate of $1 \times 10^{-6}$, weight decay of 0.01, and the AdamW optimizer. The prompt batch size is set to 256. Training proceeds for 500 steps for math reasoning, code reasoning, and LLM with search engine tasks. While for logical and medical reasoning, we train for 300 steps due to their faster convergence. The KL penalty coefficient in the objective function of GRPO is set to 0.0001, and we set the entropy coefficient as 0.001 to encourage exploration. All the rewards are derived from rule-based outcome verifiers. For code reasoning, we set the pass rate over all test cases ($\frac{\#\text{passed test cases}}{\#\text{total test cases}}$) as the reward. For other tasks, a binary reward is assigned based on the correctness of the final answer: the reward is set to 1 if the final answer is correct, otherwise, the reward is set to 0.

**Training Dataset.** The training datasets we used are listed as follows: (1) *Math Reasoning:* We train the DeepSeek-R1-Distill-Qwen-1.5B model on a combination of AIME (1984-2023), AMC problems (prior to 2023), and Omni-MATH (Gao et al., 2024) datasets. Other models are trained on 8K problems from the MATH (Hendrycks et al., 2021) dataset with difficulty levels between 3 and 5. (2) *Code Reasoning:* All models are trained on 2K programming problems from the Leet-CodeDataset (Xia et al., 2025). (3) *Medical Reasoning:* All models are trained on 10K English questions from the training split of MedQA (USMLE) (Jin et al., 2021). (4) *Logical Reasoning:* All models are trained on 3K examples from the training split of RiddleSense (Lin et al., 2021). (5) *QA with Search Engine:* All models are trained on 90K questions from the training set of HotPotQA (Yang et al., 2018).

**Evaluation Protocol.** We employ greedy decoding for all test data and report the pass@1 metric across the following benchmarks: (1) For *Math Reasoning:* we evaluate the mathematics problem-solving ability on five widely-used challenging benchmarks, including AIME24 (AIME, 2024), AMC23 (AMC, 2023), MATH-500 (Hendrycks et al., 2021), MinervaMath (Lewkowycz et al., 2022), and OlympiadBench (He et al., 2024). (2) For *Code Reasoning:* We evaluate the complex programming ability on five code generation benchmarks, including MBPP (Austin et al., 2021), MBPP+ (Liu et al., 2023), HumanEval (Chen et al., 2021), HumanEval+ (Liu et al., 2023), and LiveCodeBench (Jain et al., 2024). (3) For *Logical Reasoning:* We evaluate the LLMs' ability to solve puzzles on the well-known RiddleSense benchmark. (4) For *Medical Reasoning:* The medical reasoning evaluation is conducted using the test split of MedQA (USMLE) (Jin et al., 2021). (5) *QA with Search Engine:* We evaluate on two popular QA benchmarks, one Natural Questions (NQ) (Kwiatkowski et al., 2019) designed for general question answering, and another is HotPotQA (Yang et al., 2018), which qualifies the multi-hop reasoning ability.

**Baseline Models.** We conduct experiments across a diverse set of popular LLMs, including both the *Qwen2.5* (Yang et al., 2024a) and *LLaMA3.2* (AI, 2024) families with varying sizes (1.5B, 3B, and 7B parameters). For math and code reasoning, we additionally experiment with specialized models, *Qwen2.5-Math* (Yang et al., 2024b) and *Qwen2.5-Coder* (Hui et al., 2024), respectively. Beyond

them, we also consider a LongCoT instruction-tuned model, *DeepSeek-R1-Distill-Qwen-1.5B* (Guo et al., 2025), to examine the effectiveness of our method on the model with the recent advanced long chain-of-thought reasoning paradigm. Both base LLMs and instruction-tuned LLMs are considered to provide comprehensive comparisons on two common RL settings: (1) Directly RL starting from a base (pretrained-only) model, *i.e.,* Zero-RL, and (2) Applying RL for an instruction-tuned (SFT) model.

## C    RELATED WORK

**LLM Reasoning.** Recent research (OpenAI, 2024; 2025a;b; Guo et al., 2025) has demonstrated that large language models can achieve significant performance gains by incorporating step-by-step reasoning. Most of these approaches rely on prompting to guide the model into generating explicit and sequential reasoning paths (Wei et al., 2022; Yao et al., 2023; Besta et al., 2024). This includes well-known methods such as Chain-of-Thought (CoT) prompting (Wei et al., 2022; Kojima et al., 2022; Reynolds & McDonell, 2021; Zelikman et al., 2022; Ye et al., 2025), Tree-of-Thought (Yao et al., 2023), and Graph of Thoughts (Besta et al., 2024). Additionally, some works integrate more sophisticated search algorithms (Feng et al., 2024; Xin et al., 2024; Trinh et al., 2024) with the reasoning process, for instance, by using Monte Carlo Tree Search (Trinh et al., 2024). However, reasoning capabilities triggered solely by prompting can be unstable and may not fully unlock a model's true potential. To address this limitation, other research has proposed training-based methods to further strengthen a model's step-by-step reasoning ability. These efforts often involve creating datasets (Muennighoff et al., 2025; Min et al., 2024; Luo et al., 2025) with reasoning annotations or utilizing bootstrapping self-training techniques (Zelikman et al., 2022). To improve reasoning performance, some studies have trained verifiers to check the validity of intermediate steps (Lightman et al., 2023; Cobbe et al., 2021). The use of these verifiers has been shown to significantly boost a model's overall reasoning ability.

**Reinforcement Learning for LLMs.** A key driver of recent advancements in large language models (LLMs) (OpenAI, 2024; 2025a;b; Guo et al., 2025; Team et al., 2025) has been Reinforcement Fine-Tuning (RFT) (Schulman et al., 2017; Shao et al., 2024a; Guo et al., 2025; Yu et al., 2025; Hu, 2025), a technique that refines model behavior through reward-guided optimization. This approach fundamentally differs from Supervised Fine-Tuning (SFT) (Muennighoff et al., 2025; Min et al., 2024; Luo et al., 2025), which aligns model outputs with labeled responses, by using reinforcement learning to adapt models based on feedback signals. Typically, RFT relies on reinforcement learning algorithms like Proximal Policy Optimization (PPO) (Schulman et al., 2017) paired with rule-based reward functions. For example, DeepSeek-R1 (Guo et al., 2025) utilized Group Relative Policy Optimization (GRPO) (Shao et al., 2024a) with binary rewards to indicate the correctness of answers in tasks such as mathematics (AIME, 2024) and coding (Jain et al., 2024), achieving impressive results. A number of studies suggest that RFT not only enhances cognitive abilities like reflection and self-correction (Gandhi et al., 2025; Guo et al., 2025) but also improves generalization across various tasks (Chu et al., 2025a). Current RFT research is largely centered on algorithmic improvements. For instance, VinePPO (Kazemnejad et al., 2024) was designed to address the limitations of PPO's value networks in complex reasoning tasks by introducing unbiased Monte Carlo estimates for better credit assignment, leading to gains in both efficiency and performance over PPO baselines. Similarly, Liu et al. (2025) analyzed the training pipeline of DeepSeek-R1-Zero (Guo et al., 2025), identifying biases in GRPO and proposing Dr.GRPO to enhance both token efficiency and final performance. Other efforts have aimed to simplify the GRPO algorithm, for example, by removing the KL-divergence term to produce more robust empirical results (Yu et al., 2025; Chu et al., 2025b).

## D    ANALYSIS ON THE ADVANTAGE ESTIMATION DIFFERENCE BETWEEN MS-GRPO AND GRPO

In this section, we first provide a detailed derivation for the advantage difference $\Delta \hat{A}_{i,t}$ between Multi-Scale Group Relative Policy Optimization (MS-GRPO) and Group Relative Policy Optimization (GRPO) in Sec. D.1. Then, we prove the properties of two critical terms of $\Delta \hat{A}_{i,t}$ in Sec. D.2 and Sec. D.3.

## D.1 DERIVATION OF THE APPROXIMATE ADVANTAGE DIFFERENCE

Let $\hat{A}_{i,t}^{\text{MS-GRPO}}$ and $\hat{A}_{i,t}^{\text{GRPO}}$ denote the advantage estimates from MS-GRPO and GRPO for $o_{i,t}$ (the $t$-th token of $i$-th response) with reward $r_{i,t}$, respectively. The difference is defined as:

$$\Delta \hat{A}_{i,t} = \hat{A}_{i,t}^{\text{MS-GRPO}} - \hat{A}_{i,t}^{\text{GRPO}} \tag{11}$$

For clarity in the following formulas, we use the simplified notation $\mathbb{E}_{\mathcal{S}}[\cdot]$ to represent the hierarchical aggregation of MS-GRPO. Substituting the definitions of advantage in Eq. (6) and Eq. (1), we can expand the expression as follows:

$$
\begin{aligned}
\Delta \hat{A}_{i,t} &= \mathbb{E}_{\mathcal{S}}\left[\frac{r_{i,t} - \mu_{\mathcal{S}}}{\sigma_{\mathcal{S}}}\right] - \frac{r_{i,t} - \mu_{\mathcal{O}}}{\sigma_{\mathcal{O}}} \\
&= \mathbb{E}_{\mathcal{S}}\left[\frac{r_{i,t}}{\sigma_{\mathcal{S}}}\right] - \mathbb{E}_{\mathcal{S}}\left[\frac{\mu_{\mathcal{S}}}{\sigma_{\mathcal{S}}}\right] - \frac{r_{i,t}}{\sigma_{\mathcal{O}}} + \frac{\mu_{\mathcal{O}}}{\sigma_{\mathcal{O}}} \\
&= r_{i,t} \mathbb{E}_{\mathcal{S}}\left[\frac{1}{\sigma_{\mathcal{S}}}\right] - \mathbb{E}_{\mathcal{S}}\left[\frac{\mu_{\mathcal{S}}}{\sigma_{\mathcal{S}}}\right] - \frac{r_{i,t}}{\sigma_{\mathcal{O}}} + \frac{\mu_{\mathcal{O}}}{\sigma_{\mathcal{O}}}
\end{aligned} \tag{12}
$$

To make this expression tractable, we further derive the expressions for the two expectation terms $\mathbb{E}_{\mathcal{S}}\left[\frac{1}{\sigma_{\mathcal{S}}}\right]$ and $\mathbb{E}_{\mathcal{S}}\left[\frac{\mu_{\mathcal{S}}}{\sigma_{\mathcal{S}}}\right]$:

- **Derivation for $\mathbb{E}_{\mathcal{S}}\left[\frac{1}{\sigma_{\mathcal{S}}}\right]$:** We apply the second-order Taylor expansion on function $f(\sigma_{\mathcal{S}}) = 1/\sigma_{\mathcal{S}}$ around the mean $\mathbb{E}_{\mathcal{S}}[\sigma_{\mathcal{S}}]$:

$$f(\sigma_{\mathcal{S}}) \approx f(\mathbb{E}_{\mathcal{S}}[\sigma_{\mathcal{S}}]) + (\sigma_{\mathcal{S}} - \mathbb{E}_{\mathcal{S}}[\sigma_{\mathcal{S}}]) f'(\mathbb{E}_{\mathcal{S}}[\sigma_{\mathcal{S}}]) + \frac{1}{2}(\sigma_{\mathcal{S}} - \mathbb{E}_{\mathcal{S}}[\sigma_{\mathcal{S}}])^2 f''(\mathbb{E}_{\mathcal{S}}[\sigma_{\mathcal{S}}]) \tag{13}$$

Taking the expectation of both sides, the linear term vanishes since $\mathbb{E}_{\mathcal{S}}[\sigma_{\mathcal{S}} - \mathbb{E}_{\mathcal{S}}[\sigma_{\mathcal{S}}]] = 0$, and we get the approximation for the expectation:

$$
\begin{aligned}
\mathbb{E}_{\mathcal{S}}\left[\frac{1}{\sigma_{\mathcal{S}}}\right] &\approx \frac{1}{\mathbb{E}_{\mathcal{S}}[\sigma_{\mathcal{S}}]} + \frac{1}{2}\text{Var}_{\mathcal{S}}(\sigma_{\mathcal{S}}) \cdot f''(\mathbb{E}_{\mathcal{S}}[\sigma_{\mathcal{S}}]) \\
&= \frac{1}{\mathbb{E}_{\mathcal{S}}[\sigma_{\mathcal{S}}]} + \frac{1}{2}\text{Var}_{\mathcal{S}}(\sigma_{\mathcal{S}}) \cdot \frac{2}{(\mathbb{E}_{\mathcal{S}}[\sigma_{\mathcal{S}}])^3} \\
&= \frac{1}{\mathbb{E}_{\mathcal{S}}[\sigma_{\mathcal{S}}]} + \frac{\text{Var}_{\mathcal{S}}(\sigma_{\mathcal{S}})}{(\mathbb{E}_{\mathcal{S}}[\sigma_{\mathcal{S}}])^3}
\end{aligned} \tag{14}
$$

This result shows how the expectation of the inverse is related to the inverse of the expectation plus a correction term that is directly proportional to the variance of the random variable.

- **Derivation for $\mathbb{E}_{\mathcal{S}}\left[\frac{\mu_{\mathcal{S}}}{\sigma_{\mathcal{S}}}\right]$:** We apply the first-order multivariate Taylor expansion on function $f(\mu_{\mathcal{S}}, \sigma_{\mathcal{S}}) = \mu_{\mathcal{S}}/\sigma_{\mathcal{S}}$, giving:

$$
\begin{aligned}
f(\mu_{\mathcal{S}}, \sigma_{\mathcal{S}}) \approx {}& f(\mathbb{E}_{\mathcal{S}}[\mu_{\mathcal{S}}], \mathbb{E}_{\mathcal{S}}[\sigma_{\mathcal{S}}]) + (\mu_{\mathcal{S}} - \mathbb{E}_{\mathcal{S}}[\mu_{\mathcal{S}}]) f'_{\mu_{\mathcal{S}}}(\mathbb{E}_{\mathcal{S}}[\mu_{\mathcal{S}}], \mathbb{E}_{\mathcal{S}}[\sigma_{\mathcal{S}}]) \\
&+ (\sigma_{\mathcal{S}} - \mathbb{E}_{\mathcal{S}}[\sigma_{\mathcal{S}}]) f'_{\sigma_{\mathcal{S}}}(\mathbb{E}_{\mathcal{S}}[\mu_{\mathcal{S}}], \mathbb{E}_{\mathcal{S}}[\sigma_{\mathcal{S}}])
\end{aligned} \tag{15}
$$

Taking the expectation of both sides, the linear terms vanish since $\mathbb{E}_{\mathcal{S}}[\mu_{\mathcal{S}} - \mathbb{E}_{\mathcal{S}}[\mu_{\mathcal{S}}]] = 0$ and $\mathbb{E}_{\mathcal{S}}[\sigma_{\mathcal{S}} - \mathbb{E}_{\mathcal{S}}[\sigma_{\mathcal{S}}]] = 0$, and $\mathbb{E}_{\mathcal{S}}[\mu_{\mathcal{S}}] = \mu_{\mathcal{O}}$ (given by the Law of Total Expectation), we get the approximation for the expectation:

$$\mathbb{E}_{\mathcal{S}}\left[\frac{\mu_{\mathcal{S}}}{\sigma_{\mathcal{S}}}\right] \approx \frac{\mathbb{E}_{\mathcal{S}}[\mu_{\mathcal{S}}]}{\mathbb{E}_{\mathcal{S}}[\sigma_{\mathcal{S}}]} = \frac{\mu_{\mathcal{O}}}{\mathbb{E}_{\mathcal{S}}[\sigma_{\mathcal{S}}]} \tag{16}$$

Now, we substitute the expressions of the two expectation terms in Eq. (14) and Eq. (16) back into the expanded difference formula in Eq. (12). Plugging in the results from the derivations above and grouping the terms containing $r_{i,t}$ and $\mu_{\mathcal{O}}$, we have:

$$
\begin{aligned}
\Delta \hat{A}_{i,t} &\approx r_{i,t}\left(\frac{1}{\mathbb{E}_{\mathcal{S}}[\sigma_{\mathcal{S}}]} + \frac{\text{Var}_{\mathcal{S}}(\sigma_{\mathcal{S}})}{(\mathbb{E}_{\mathcal{S}}[\sigma_{\mathcal{S}}])^3}\right) - \left(\frac{\mu_{\mathcal{O}}}{\mathbb{E}_{\mathcal{S}}[\sigma_{\mathcal{S}}]}\right) - \frac{r_{i,t}}{\sigma_{\mathcal{O}}} + \frac{\mu_{\mathcal{O}}}{\sigma_{\mathcal{O}}} \\
&= r_{i,t}\left(\frac{1}{\mathbb{E}_{\mathcal{S}}[\sigma_{\mathcal{S}}]} - \frac{1}{\sigma_{\mathcal{O}}}\right) - \mu_{\mathcal{O}}\left(\frac{1}{\mathbb{E}_{\mathcal{S}}[\sigma_{\mathcal{S}}]} - \frac{1}{\sigma_{\mathcal{O}}}\right) + r_{i,t}\frac{\text{Var}_{\mathcal{S}}(\sigma_{\mathcal{S}})}{(\mathbb{E}_{\mathcal{S}}[\sigma_{\mathcal{S}}])^3} \\
&= (r_{i,t} - \mu_{\mathcal{O}})\left(\frac{1}{\mathbb{E}_{\mathcal{S}}[\sigma_{\mathcal{S}}]} - \frac{1}{\sigma_{\mathcal{O}}}\right) + r_{i,t}\frac{\text{Var}_{\mathcal{S}}(\sigma_{\mathcal{S}})}{(\mathbb{E}_{\mathcal{S}}[\sigma_{\mathcal{S}}])^3}
\end{aligned} \tag{17}
$$

To analyze the terms above, we need to find an expression for the remaining expectation term $\mathbb{E}_{\mathcal{S}}[\sigma_{\mathcal{S}}]$. Next, we continue to derive it:

- **Derivation for $\mathbb{E}_{\mathcal{S}}[\sigma_{\mathcal{S}}]$:** First, we establish a relationship between the expectation of the squared local standard deviation, $\mathbb{E}_{\mathcal{S}}[\sigma_{\mathcal{S}}^2]$, and the square of its expectation, $(\mathbb{E}_{\mathcal{S}}[\sigma_{\mathcal{S}}])^2$. This identity is derived directly from the fundamental definition of variance:

$$
\begin{aligned}
\text{Var}(\sigma_{\mathcal{S}}) &= \mathbb{E}_{\mathcal{S}}\left[(\sigma_{\mathcal{S}} - \mathbb{E}_{\mathcal{S}}[\sigma_{\mathcal{S}}])^2\right] \\
&= \mathbb{E}_{\mathcal{S}}\left[\sigma_{\mathcal{S}}^2 - 2\sigma_{\mathcal{S}}\mathbb{E}_{\mathcal{S}}[\sigma_{\mathcal{S}}] + (\mathbb{E}_{\mathcal{S}}[\sigma_{\mathcal{S}}])^2\right] \\
&= \mathbb{E}_{\mathcal{S}}[\sigma_{\mathcal{S}}^2] - \mathbb{E}_{\mathcal{S}}\left[2\sigma_{\mathcal{S}}\mathbb{E}_{\mathcal{S}}[\sigma_{\mathcal{S}}]\right] + \mathbb{E}_{\mathcal{S}}\left[(\mathbb{E}_{\mathcal{S}}[\sigma_{\mathcal{S}}])^2\right] \\
&= \mathbb{E}_{\mathcal{S}}[\sigma_{\mathcal{S}}^2] - 2\mathbb{E}_{\mathcal{S}}[\sigma_{\mathcal{S}}]\mathbb{E}[\sigma_{\mathcal{S}}] + (\mathbb{E}_{\mathcal{S}}[\sigma_{\mathcal{S}}])^2 \\
&= \mathbb{E}_{\mathcal{S}}[\sigma_{\mathcal{S}}^2] - (\mathbb{E}_{\mathcal{S}}[\sigma_{\mathcal{S}}])^2
\end{aligned}
\tag{18}
$$

Rearranging this identity gives the desired relationship:

$$
(\mathbb{E}_{\mathcal{S}}[\sigma_{\mathcal{S}}])^2 = \mathbb{E}_{\mathcal{S}}[\sigma_{\mathcal{S}}^2] - \text{Var}(\sigma_{\mathcal{S}})
\tag{19}
$$

Then, according the *Law of Total Variance*, we have the expression for $\mathbb{E}_{\mathcal{S}}[\sigma_{\mathcal{S}}^2]$:

$$
\mathbb{E}_{\mathcal{S}}[\sigma_{\mathcal{S}}^2] = \sigma_{\mathcal{O}}^2 - \text{Var}_{\mathcal{S}}(\mu_{\mathcal{S}})
\tag{20}
$$

By equating Eq. (19) and Eq. (20), we can solve for $(\mathbb{E}_{\mathcal{S}}[\sigma_{\mathcal{S}}])^2$:

$$
(\mathbb{E}_{\mathcal{S}}[\sigma_{\mathcal{S}}])^2 = \sigma_{\mathcal{O}}^2 - \text{Var}_{\mathcal{S}}(\mu_{\mathcal{S}}) - \text{Var}_{\mathcal{S}}(\sigma_{\mathcal{S}})
\tag{21}
$$

Taking the square root gives us the desired expression for the average local standard deviation:

$$
\mathbb{E}_{\mathcal{S}}[\sigma_{\mathcal{S}}] = \sqrt{\sigma_{\mathcal{O}}^2 - \text{Var}_{\mathcal{S}}(\mu_{\mathcal{S}}) - \text{Var}_{\mathcal{S}}(\sigma_{\mathcal{S}})}
\tag{22}
$$

Substituting this back into Eq. (17) gives its exact expression, forming the basis of our analysis in the following subsection.

$$
\Delta \hat{A}_{i,t} \approx \underbrace{(r_{i,t} - \mu_{\mathcal{O}})}_{\text{global-relative reward}} \overbrace{\underbrace{\left(\frac{1}{\mathbb{E}_{\mathcal{S}}[\sigma_{\mathcal{S}}]} - \frac{1}{\sigma_{\mathcal{O}}}\right)}_{\text{scaling factor 1}}}^{\text{Correction Term 1}} + \underbrace{r_{i,t}}_{\text{reward}} \overbrace{\underbrace{\frac{\text{Var}_{\mathcal{S}}(\sigma_{\mathcal{S}})}{(\mathbb{E}_{\mathcal{S}}[\sigma_{\mathcal{S}}])^3}}_{\text{scaling factor 2}}}^{\text{Correction Term 2}},
\tag{23}
$$

$$
\text{where } \mathbb{E}_{\mathcal{S}}[\sigma_{\mathcal{S}}] = \sqrt{\sigma_{\mathcal{O}}^2 - \text{Var}_{\mathcal{S}}(\mu_{\mathcal{S}}) - \text{Var}_{\mathcal{S}}(\sigma_{\mathcal{S}})}
$$

### D.2 ANALYSIS ON THE PROPERTIES OF "SCALING FACTOR 1" IN EQ. (23)

The "scaling factor 1" in Eq. (23) has two crucial properties:
- It is non-negative.
- Its magnitude increases with the heterogeneity of reward distribution.

Below, we prove these properties in Proof 1 and Proof 2, respectively.

**Proof 1: The "scaling factor 1"** $\left(\mathbb{E}_{\mathcal{S}}\left[\frac{1}{\sigma_{\mathcal{S}}}\right] - \frac{1}{\sigma_{\mathcal{O}}}\right)$ **is non-negative**

This property can be formally proven using the expression for $\mathbb{E}_{\mathcal{S}}[\sigma_{\mathcal{S}}]$ derived in Eq. (23), *i.e.,* $\mathbb{E}_{\mathcal{S}}[\sigma_{\mathcal{S}}] = \sqrt{\sigma_{\mathcal{O}}^2 - \text{Var}_{\mathcal{S}}(\mu_{\mathcal{S}}) - \text{Var}_{\mathcal{S}}(\sigma_{\mathcal{S}})}$. Since both variance terms, $\text{Var}_{\mathcal{S}}(\mu_{\mathcal{S}})$ and $\text{Var}_{\mathcal{S}}(\sigma_{\mathcal{S}})$, are non-negative by definition, the term inside the square root is always less than or equal to $\sigma_{\mathcal{O}}^2$. This leads to the inequality $\mathbb{E}_{\mathcal{S}}[\sigma_{\mathcal{S}}] \leq \sigma_{\mathcal{O}}$. Taking the reciprocal of both sides reverses the inequality, which in turn proves that: $\mathbb{E}_{\mathcal{S}}\left[\frac{1}{\sigma_{\mathcal{S}}}\right] - \frac{1}{\sigma_{\mathcal{O}}}$.

**Proof 2: The magnitude of "scaling factor 1"** $\left(\mathbb{E}_{\mathcal{S}}\left[\frac{1}{\sigma_{\mathcal{S}}}\right] - \frac{1}{\sigma_{\mathcal{O}}}\right)$ **increases with the heterogeneity of reward distribution.**

The concept of *reward heterogeneity* refers to the degree of non-uniformity in the reward distribution within group $\mathcal{O}$. High heterogeneity implies an uneven spread of rewards, potentially characterized by significant outliers or the emergence of distinct clusters, such as a "high-reward" subgroup or a "low-reward" subgroup. Reward heterogeneity can be quantitatively captured by two key terms: $\text{Var}_\mathcal{S}(\mu_\mathcal{S})$ and $\text{Var}_\mathcal{S}(\sigma_\mathcal{S})$.

- First, $\text{Var}_\mathcal{S}(\mu_\mathcal{S})$, the variance of subgroup means, serves as the primary and most direct measure of heterogeneity. As revealed by the Law of Total Variance ($\sigma_\mathcal{O}^2 = \mathbb{E}_\mathcal{S}[\sigma_\mathcal{S}^2] + \text{Var}_\mathcal{S}(\mu_\mathcal{S})$), $\text{Var}_\mathcal{S}(\mu_\mathcal{S})$ accounts for the portion of global variance attributable to differences between subgroup averages. It thus reflects how much the "center of gravity" of rewards shifts across local views. A large $\text{Var}_\mathcal{S}(\mu_\mathcal{S})$ is a strong indicator of clustered reward structures.

- Second, $\text{Var}_\mathcal{S}(\sigma_\mathcal{S})$, the variance of subgroup standard deviations, acts as a secondary but informative signature of heterogeneity. In heterogeneous populations, subgroups exhibit highly variable internal dispersion, for instance, a homogeneous high-reward subgroup will have low $\sigma_\mathcal{S}$, while a mixed subgroup may display high variability. The fluctuation in these local standard deviations across subgroups is precisely what $\text{Var}_\mathcal{S}(\sigma_\mathcal{S})$ measures.

In conclusion, as reward heterogeneity increases, $\text{Var}_\mathcal{S}(\mu_\mathcal{S})$ and $\text{Var}_\mathcal{S}(\sigma_\mathcal{S})$ grow, widening the gap between $\sigma_\mathcal{O}^2$ and $\mathbb{E}_\mathcal{S}[\sigma_\mathcal{S}^2]$, and thereby increasing the magnitude of $(1/\mathbb{E}_\mathcal{S}[\sigma_\mathcal{S}] - 1/\sigma_\mathcal{O})$. This establishes a clear link between structural heterogeneity in rewards and the magnitude of the second factor $(1/\mathbb{E}_\mathcal{S}[\sigma_\mathcal{S}] - 1/\sigma_\mathcal{O})$.

### D.3   ANALYSIS ON THE PROPERTIES OF "SCALING FACTOR 2" IN EQ. (23)

> The "scaling factor 2" in Eq. (23) has two crucial properties:
> - It is non-negative.
> - Its magnitude increases with the heterogeneity of reward distribution.

Below, we prove these properties in Proof 3 and Proof 4, respectively.

**Proof 3: The "scaling factor 2"** $\left(\frac{\text{Var}_\mathcal{S}(\sigma_\mathcal{S})}{(\mathbb{E}_\mathcal{S}[\sigma_\mathcal{S}])^3}\right)$ **is non-negative**

First, the numerator term $\text{Var}_\mathcal{S}(\sigma_\mathcal{S})$ represents the variance of the random variable $\sigma_\mathcal{S}$. By the definition of variance, $\text{Var}_\mathcal{S}(\sigma_\mathcal{S})$ is always non-negative, *i.e.,* $\text{Var}_\mathcal{S}(\sigma_\mathcal{S}) \geq 0$.

Second, for the denominator term $(\mathbb{E}_\mathcal{S}[\sigma_\mathcal{S}])^3$, to determine the sign, we first analyze its base, $\mathbb{E}_\mathcal{S}[\sigma_\mathcal{S}]$. Taking the expression from Eq. (23), we can know that $\mathbb{E}_\mathcal{S}[\sigma_\mathcal{S}] = \sqrt{\sigma_\mathcal{O}^2 - \text{Var}_\mathcal{S}(\mu_\mathcal{S}) - \text{Var}_\mathcal{S}(\sigma_\mathcal{S})} \geq 0$. Thus, its cube, $(\mathbb{E}_\mathcal{S}[\sigma_\mathcal{S}])^3$, must also be non-negative, *i.e.,* $(\mathbb{E}_\mathcal{S}[\sigma_\mathcal{S}])^3 \geq 0$.

In conclusion, since both the numerator $\text{Var}_\mathcal{S}(\sigma_\mathcal{S})$ and denominator $(\mathbb{E}_\mathcal{S}[\sigma_\mathcal{S}])^3$ are non-negative, their ratio is guaranteed to be non-negative. We have proven that $\frac{\text{Var}_\mathcal{S}(\sigma_\mathcal{S})}{(\mathbb{E}_\mathcal{S}[\sigma_\mathcal{S}])^3} \geq 0$.

**Proof 4: The magnitude of "scaling factor 2"** $\left(\frac{\text{Var}_\mathcal{S}(\sigma_\mathcal{S})}{(\mathbb{E}_\mathcal{S}[\sigma_\mathcal{S}])^3}\right)$ **increases with the heterogeneity of reward distribution.**

As discussed in Proof 2, the reward heterogeneity can be quantitatively measured by terms such as $\text{Var}_\mathcal{S}(\mu_\mathcal{S})$ and $\text{Var}_\mathcal{S}(\sigma_\mathcal{S})$. As reward heterogeneity increases, $\text{Var}_\mathcal{S}(\mu_\mathcal{S})$ and $\text{Var}_\mathcal{S}(\sigma_\mathcal{S})$ also grows.

Recalling the expression of $\mathbb{E}_\mathcal{S}[\sigma_\mathcal{S}]$ derived in Eq. (23), we have $\mathbb{E}_\mathcal{S}[\sigma_\mathcal{S}] = \sqrt{\sigma_\mathcal{O}^2 - \text{Var}_\mathcal{S}(\mu_\mathcal{S}) - \text{Var}_\mathcal{S}(\sigma_\mathcal{S})}$. As reward heterogeneity increases, $\text{Var}_\mathcal{S}(\mu_\mathcal{S})$ and $\text{Var}_\mathcal{S}(\sigma_\mathcal{S})$ also grows, thereby leading to the corresponding decrease of $\mathbb{E}_\mathcal{S}[\sigma_\mathcal{S}]$. Since $\mathbb{E}_\mathcal{S}[\sigma_\mathcal{S}] > 0$, its cube, $(\mathbb{E}_\mathcal{S}[\sigma_\mathcal{S}])^3$ decreases as well. Thus, as the reward heterogeneity increases, the numerator $\text{Var}_\mathcal{S}(\sigma_\mathcal{S})$ increases, while the denominator $(\mathbb{E}_\mathcal{S}[\sigma_\mathcal{S}])^3$ declines, thereby resulting in a larger $\frac{\text{Var}_\mathcal{S}(\sigma_\mathcal{S})}{(\mathbb{E}_\mathcal{S}[\sigma_\mathcal{S}])^3}$. This formally proves that the magnitude of "scaling factor 2", $\frac{\text{Var}_\mathcal{S}(\sigma_\mathcal{S})}{(\mathbb{E}_\mathcal{S}[\sigma_\mathcal{S}])^3}$, is a increasing function of the heterogeneity of the reward distribution.

# E APPROXIMATION GUARANTEE FOR THE GREEDY ALGORITHM

In this section, we provide a theoretical justification for the greedy algorithm used for the *Diverse Group Sampling* procedure (Sec. 3.2.2). We first formalize the problem and then provide a proof sketch for the approximation guarantee in the context of submodular function maximization, to which our diversity objective is closely related.

## E.1 PROBLEM FORMULATION

As stated in the main text, for a given scale $\tau$, we aim to select a subset of comparison groups $\mathbb{C}_\tau$ of size $K$ from an exhaustive set $\mathbb{S}_\tau$ that solves the Maximum Diversity problem:

$$\mathbb{C}_\tau^* = \arg\max_{\mathbb{C}_\tau \subset \mathbb{S}_\tau, |\mathbb{C}_\tau|=K} \sum_{\mathcal{S}_a, \mathcal{S}_b \in \mathbb{C}_\tau, a \neq b} d_J(\mathcal{S}_a, \mathcal{S}_b). \tag{24}$$

Let $f(\mathbb{C}_\tau) = \sum_{\mathcal{S}_a, \mathcal{S}_b \in \mathbb{C}_\tau, a \neq b} d_J(\mathcal{S}_a, \mathcal{S}_b)$ be our objective function. As this problem is NP-hard, we employ a greedy algorithm that iteratively constructs the set. Let $\mathbb{C}_k$ be the set of $k$ groups selected after $k$ iterations. In step $k+1$, the algorithm selects the group $\mathcal{S}_{k+1}$ that provides the maximum marginal gain:

$$\mathcal{S}_{k+1} = \arg\max_{\mathcal{S} \in \mathbb{S}_\tau \setminus \mathbb{C}_k} f(\mathbb{C}_k \cup \{\mathcal{S}\}) - f(\mathbb{C}_k). \tag{25}$$

Note that the selection criterion in the main text, $\sum_{\mathcal{S}' \in \mathbb{C}_k} d_J(\mathcal{S}, \mathcal{S}')$, is precisely this marginal gain.

## E.2 SUBMODULARITY AND APPROXIMATION GUARANTEES

The effectiveness of this greedy strategy is best understood through the lens of *submodularity*. A set function $f : 2^V \to \mathbb{R}$ is submodular if for any two sets $A \subseteq B \subset V$ and any element $x \in V \setminus B$, it satisfies the "diminishing returns" property:

$$f(A \cup \{x\}) - f(A) \geq f(B \cup \{x\}) - f(B). \tag{26}$$

In words, the marginal gain of adding an element $x$ to a small set $A$ is greater than or equal to the marginal gain of adding the same element to a larger superset $B$.

While our Max-Sum Diversity objective $f(\mathbb{C}_\tau)$ is not strictly submodular, it is closely related to this class of functions, and the greedy algorithm is the principled approach for both. For the general class of non-negative, monotone submodular functions, a celebrated result provides a strong performance guarantee for the greedy algorithm.

The theorem in (Nemhauser et al., 1978) states that: for the problem of maximizing a non-negative, monotone submodular function $f$ subject to a cardinality constraint $|\mathbb{C}| \leq K$, the greedy algorithm produces a solution $\mathbb{C}_G$ such that: $f(\mathbb{C}_G) \geq \left(1 - \frac{1}{e}\right) f(\mathbb{C}^*)$, where $\mathbb{C}^*$ is the true optimal solution and $e$ is the base of the natural logarithm. This guarantees that the greedy solution is within approximately 63.2% of the optimal solution.

## E.3 PROOF SKETCH FOR THE SUBMODULAR CASE

We provide a sketch of the classic proof for the $(1 - 1/e)$ approximation guarantee to illustrate the principle. Let $\mathbb{C}_k$ be the greedy set of size $k$ and $\mathbb{C}^*$ be the optimal set of size $K$. Let $g_k = f(\mathbb{C}_k) - f(\mathbb{C}_{k-1})$ be the marginal gain at step $k$.

The key insight is to bound the difference between the optimal value and the current greedy value. By monotonicity and submodularity, we can write:

$$f(\mathbb{C}^*) \leq f(\mathbb{C}_k \cup \mathbb{C}^*) \tag{27}$$

$$= f(\mathbb{C}_k) + \sum_{x \in \mathbb{C}^* \setminus \mathbb{C}_k} (f(\mathbb{C}_k \cup \{x\}) - f(\mathbb{C}_k)) \tag{28}$$

$$\leq f(\mathbb{C}_k) + \sum_{x \in \mathbb{C}^* \setminus \mathbb{C}_k} (f(\mathbb{C}_{k-1} \cup \{x\}) - f(\mathbb{C}_{k-1})) \quad \text{(by submodularity)} \tag{29}$$

The greedy choice at step $k$ maximizes the marginal gain, so its gain $g_k$ is at least the average gain of the elements in $\mathbb{C}^* \setminus \mathbb{C}_{k-1}$. This leads to the inequality:

$$f(\mathbb{C}^*) - f(\mathbb{C}_{k-1}) \leq K \cdot g_k = K \cdot (f(\mathbb{C}_k) - f(\mathbb{C}_{k-1})) \tag{30}$$

Rearranging this recurrence relation over $k$ from 1 to $K$ and using the fact that $(1 - 1/K)^K \approx 1/e$ leads to the final bound $f(\mathbb{C}_K) \geq (1 - 1/e)f(\mathbb{C}^*)$.

This proof sketch demonstrates how the diminishing returns property of submodularity allows the greedy algorithm to provide a constant-factor approximation of the true optimum. While our specific objective requires a more tailored analysis, this classic result provides the theoretical foundation for why the greedy approach is a fast, principled, and provably effective choice for our diversity maximization task.

