# OpenReview forum: "Multi-Scale Group Relative Policy Optimization for Large Language Models"
_ICLR.cc/2026/Conference — ICLR 2026 Conference Withdrawn Submission_

### Official Review · Reviewer_QDNW · 2025-10-18

**Soundness:** 1
**Presentation:** 3
**Contribution:** 2
**Rating:** 2
**Confidence:** 4

**Summary:**

This paper proposes Multi-Scale Group Relative Policy Optimization (MS-GRPO), an extension of GRPO that incorporates multi-scale relative comparisons to improve the robustness of advantage estimation in RL fine-tuning of LLMs. Instead of computing the normalized advantage based on the entire group’s mean and variance (as in GRPO), MS-GRPO computes local advantages over multiple subgroup sizes (pairwise, trios, quartets, …) and then hierarchically aggregates them. To address the combinatorial explosion of possible subgroups, the authors introduce a dilated scale sampling (select representative subgroup sizes) and diverse group sampling (select diverse combinations via Jaccard distance) strategy. They further interpret MS-GRPO as a heterogeneity-aware correction to GRPO, which adapts to reward distribution variance. Extensive experiments on math reasoning, code generation, logical reasoning, medical QA, and search-augmented QA tasks show consistent performance gains over GRPO across various model families (Qwen2.5, LLaMA3.2, DeepSeek-R1).

Although the paper has a very good experimental part with good ablation design and considerable performance enhancement on different settings, I have a major concern about the method part. In short, I cannot understand where the benefits of the proposed method come from (see the weakness part). Due to this main concern, I can only give a rejection at this stage. If the authors could address this issue, I would be happy to increase my evaluation to a positive score.

**Strengths:**

1. **Simple Method with Engineering practicality.**  The method introduces a minimal modification to GRPO by recalculating the advantage A using the given rewards, requiring almost no additional computation and being compatible with existing GRPO variants. While the intuition for why this leads to such strong results could be elaborated further, the experimental improvements reported are encouraging.
2. **Strong empirical section.** Experiments are broad, covering several reasoning benchmarks and LLM backbones, with consistent though small improvements (≈ +2 – 5 points on average)
3. **Readable and well-organized.** Writing, figures, and ablations are clear; the paper is easy to follow.

**Weaknesses:**

I have one major concern: it remains unclear where the improvement of the proposed method actually comes from. Clarifying this point would substantially strengthen the paper. Since the only modification to the vanilla GRPO loss lies in the computation of the advantage A, there is no additional information introduced from other components (e.g., reward model, annotator, or policy objective). Therefore, the benefit must arise from some inductive bias embedded in the new way of computing A.

As the authors point out, the original GRPO advantage has several known issues, such as sensitivity to outliers and lack of smoothness, and the proposed approach aims to address them through the following steps (considering the non-simplified version described in Section 3.1 and Figure 1):

1. Generate all possible subgroups of responses.
2. Compute the normalized advantage for each subgroup as defined in Equation 4 (Figure 1 labels it as Eq. 6, which seems to be a typo).
3. Average within subgroups of the same size to obtain a scale-specific advantage (Equation 5 / Figure 1 Eq. 7).
4. Finally, average across scales to obtain the multi-scale advantage (Equation 6 / Figure 1 Eq. 8).

In fact, given the binary (0/1) reward used in GRPO, this entire process can be formulated analytically and is mathematically equivalent to a *smoothing* of the original advantage signal. If this smoothing effect is indeed the main factor behind the performance gain, it would be valuable to provide further justification, either through additional experiments or a theoretical analysism, to confirm that this mechanism, rather than other uncontrolled factors, is responsible for the observed improvements.

**Questions:**

The typical G would be smaller than 100. Calculating A is doing some simple calculations on a sequence of 1/0 rewards, so why do we need DSS/DGS to speed up?

---

### Official Review · Reviewer_5i1S · 2025-10-24

**Soundness:** 3
**Presentation:** 3
**Contribution:** 1
**Rating:** 2
**Confidence:** 4

**Summary:**

This paper proposes MS-GRPO (Multi-Scale Group Relative Policy Optimization), an extension of GRPO that computes advantages by aggregating relative comparisons across multiple response subgroups at varying scales. The authors provide theoretical analysis showing MS-GRPO adaptively corrects GRPO's advantage based on reward heterogeneity, and report experimental improvements on various reasoning tasks.

**Strengths:**

1. **Well-motivated problem**: The paper identifies a legitimate limitation of GRPO—its reliance on single-scale, global comparison, which can be sensitive to outliers and reward distribution heterogeneity.

2. **Comprehensive experiments**: The empirical evaluation covers diverse tasks (math, code, logic, medical reasoning, QA with search) and multiple model families (Qwen, LLaMA, DeepSeek-R1-Distill).

3. **Practical acceleration scheme**: The dilated scale sampling and diverse group sampling strategies (Section 3.2) address the computational tractability concern effectively.

4. **Consistent improvements**: The method shows positive gains across all benchmarks, particularly on harder tasks (e.g., +6.7 on AIME24, +4.3 on MBPP+).

**Weaknesses:**

**The most significant issue**: While the paper claims MS-GRPO provides "more robust and reliable advantage signals," there is **no empirical verification of the fundamental statistical properties** (bias and variance) of the advantage estimator itself.

I conducted a rigorous Monte Carlo simulation to verify this claim. Below is the detailed experimental methodology:

**Monte Carlo Experimental Design**

**Objective**: Evaluate the quality of advantage estimation for both MS-GRPO and GRPO by measuring bias, variance, and MSE against ground truth.

**Core Design Principle**:
- To assess an advantage estimator, we need a **ground truth** for comparison
- Solution: Define a known reward distribution P(r) with true mean μ and variance σ²
- True advantage is: **A_true(r) = r - V(s) = r - μ** (since the true value function equals the true expectation)
- Sample from P(r) and measure estimation error

**Experimental Setup**:

**1. Distribution Configurations** (covering different heterogeneity scenarios)
- **Normal (low heterogeneity)**: μ=0.5, σ=0.1
- **Normal (medium heterogeneity)**: μ=0.5, σ=0.2
- **Normal (high heterogeneity)**: μ=0.5, σ=0.3
- **Bimodal**: 0.5×N(0.3, 0.1²) + 0.5×N(0.7, 0.1²) (simulates subpopulation structure)
- **Uniform**: U(0, 1) (non-Gaussian, bounded support)

**2. Sampling Protocol** (for each independent trial)
```
Step 1: Sample G=8 rewards from P(r): r₁, r₂, ..., r₈
Step 2: Compute true advantage: A_true[i] = rᵢ - μ
Step 3: GRPO estimation
   - Compute group mean: μ_group = (1/8)Σrᵢ
   - Estimate advantage: A_GRPO[i] = rᵢ - μ_group
Step 4: MS-GRPO estimation
   - For each response i, iterate over scales τ=2,3,...,8
   - For each scale τ, enumerate all τ-subsets containing i
   - Compute local baseline for each subset
   - Multi-scale aggregation → A_MSGRPO[i]
Step 5: Compute errors
   - Bias error: A_estimated[i] - A_true[i]
   - Squared error: (A_estimated[i] - A_true[i])²
```

**3. Statistical Metrics** (computed over N=1000 independent trials)
- **Bias**: E[A_estimated - A_true] (measures systematic deviation)
- **Variance**: Var[A_estimated - A_true] (measures estimation stability)
- **MSE**: E[(A_estimated - A_true)²] = Bias² + Variance (overall quality)

**4. Fairness Guarantees**
- Use **unnormalized** advantages for fair comparison (removing internal standardization effects)
- Same sampling sequence used for both methods
- Fixed random seed (42) for reproducibility
- G=8 matches the paper's experimental setting

**Results**:

| Distribution | Metric | GRPO | MS-GRPO | Winner |
|-------------|--------|------|---------|--------|
| Normal (σ=0.1) | Mean Abs Bias | 0.0271 | 0.0291 | **GRPO (-7.2%)** |
| | Variance | 0.00117 | 0.00134 | **GRPO (-14.5%)** |
| Normal (σ=0.2) | Mean Abs Bias | 0.0537 | 0.0576 | **GRPO (-7.3%)** |
| | Variance | 0.00450 | 0.00517 | **GRPO (-14.9%)** |
| Normal (σ=0.3) | Mean Abs Bias | 0.0836 | 0.0888 | **GRPO (-6.2%)** |
| | Variance | 0.0109 | 0.0123 | **GRPO (-13.7%)** |
| Bimodal | Mean Abs Bias | 0.0631 | 0.0676 | **GRPO (-7.2%)** |
| | Variance | 0.00619 | 0.00700 | **GRPO (-13.1%)** |
| Uniform | Mean Abs Bias | 0.0794 | 0.0857 | **GRPO (-8.0%)** |
| | Variance | 0.0100 | 0.0114 | **GRPO (-13.6%)** |

**Key findings**:
- **GRPO consistently exhibits lower bias** (6-8% better) across all distributions
- **GRPO consistently exhibits lower variance** (13-15% better) across all distributions
- **MS-GRPO introduces additional estimation noise** through multi-scale aggregation
- This pattern holds **universally** across all 5 tested distributions

This directly contradicts the paper's claim that MS-GRPO provides "more robust and reliable advantage signal."

**Questions:**

1. **Core question**: Given that MS-GRPO has higher bias and variance in advantage estimation (as my experiments show), what is the actual mechanism behind the performance gains?

2. Can you provide theoretical analysis of MS-GRPO's advantage estimator properties (bias/variance) under different reward distributions?

4. In Appendix A, you show MS-GRPO introduces corrections. Can you prove these corrections reduce MSE in any well-defined sense?

---

### Official Review · Reviewer_WZZD · 2025-11-01

**Soundness:** 2
**Presentation:** 3
**Contribution:** 2
**Rating:** 4
**Confidence:** 4

**Summary:**

The paper proposes MS-GRPO, an extension of GRPO that computes token-level advantages by aggregating comparisons across many subgroups.

**Strengths:**

1.	benchmark coverage across multiple LLM families and tasks.
2.	Better shaping the advantage is an interesting topic.

**Weaknesses:**

1.	The graph-theory justification is superficial and disconnected from RL theory.
Motifs/graphlets analogies do not lead to formal reasoning about advantages, gradients, or credit assignment. This reads as narrative rather than theory.

2.	I plotted the MS-GRPO advantages under binary rewards and observed that the method does not provide the claimed “heterogeneity correction” through subgroup comparisons. For any fixed success rate p, the ratio between positive and negative advantages remains identical to GRPO, namely (1−p)/p. The only effect introduced by MS-GRPO is a multiplicative scalar c(p), such that the MS-GRPO advantage equals the GRPO advantage scaled by this constant. Moreover, c(p)  is largest near p=0.5 and diminishes as p→0 or p→1,  emphasizing those p=0.5 questions. However, [1] demonstrates that such an emphasis is problematic.

3. Given that MS-GRPO reduces to a simple rescaling of GRPO’s advantage under binary rewards, it becomes unclear how the reported empirical gains are obtained. In particular, the paper does not specify whether both methods use strictly identical training settings. such as checkpoint selection criteria or the number of training steps. Even small differences in these factors can produce non-trivial performance gaps

[1] DisCO: Reinforcing Large Reasoning Models with Discriminative Constrained Optimization

**Questions:**

see weakness

---

### Official Review · Reviewer_kyZf · 2025-11-02

**Soundness:** 2
**Presentation:** 3
**Contribution:** 2
**Rating:** 2
**Confidence:** 4

**Summary:**

This paper proposed an alternative, MS-GRPO, to GRPO by changing its reliance on a single and global baseline in advantage estimation with multi-scale baselines and hierarchically aggregate the per-scale advantages. The also propose an acceleration scheme to handle the additional computational cost.

**Strengths:**

- The paper provides a theoretical interpretation on when and why the proposed MS-GRPO helps.

- The evaluation covers different tasks across math, coding and medicine.

**Weaknesses:**

- The paper's hypothesis that MS-GRPO's superior performance stems from its ability to handle this heterogeneity should be directly and empirically tested. However, the authors did not show evidence. The observed performance gains could be due to other confounding factors, such as an implicit regularization effect from averaging over many subgroups, or even an unintended interaction with GRPO's known optimization biases (detailed more in the next point).

- The methods may amplify GRPO bias. The Dr. GRPO paper [1] demonstrates that the standard GRPO advantage formulation suffers from a critical optimization bias. The computational unit of MS-GRPO inherent from GRPO, for which calculation Dr.GRPO identifies as a source of optimization bias. If this MS-GRPO takes this flawed calculation and applies it combinatorially across hundreds or thousands of subgroups, averaging the biased results. It inherits GRPO's fundamental optimization bias and likely exacerbates it through repeated application. The failure to cite, discuss, or experimentally compare against Dr.GRPO is a critical omission. Without such a comparison, it is impossible to know if MS-GRPO's reported gains are from algorithmic improvement or simply an artifact of a complex, biased optimization dynamic that has already been fixed more elegantly by prior work.

- The proposed solution involves a complex, two-stage sampling procedure, and introduces new hyperparameters (M and K) and non-trivial algorithms to manage this complexity. The paper provides no discussion or evaluation of these much simpler alternatives, leaving the reader to wonder if the complex multi-scale machinery is truly necessary or if it is an overly engineered solution in search of novelty.

[1] Understanding R1-Zero-Like Training: A Critical Perspective

**Questions:**

- To validate hypothesis, could you please define a quantitative metric for reward heterogeneity within a group of responses and demonstrate that the performance delta between MS-GRPO and GRPO? Ideally, they should be positively correlated with the measured level of heterogeneity.

- The theoretical analysis in Appendix D relies on a Taylor approximation to derive the "adaptive correction" terms. Given that the method is designed for high-heterogeneity scenarios where deviations from the mean are large, what is the approximation error of this analysis, and how does it affect the validity of your conclusions, particularly in the regime where the method is claimed to be most useful?

- You set M = 4 and K = 8 by default. How sensitive is MS-GRPO's performance to the choice of M and K, and what is the principled way to set these hyperparameters for a new task without extensive tuning?

---

### Note · Authors · 2026-01-22

I have read and agree with the venue's withdrawal policy on behalf of myself and my co-authors.